# Greenland ice sheet runoff reduced by meltwater refreezing in bare ice

Matthew G. Cooper [1,2] ✉, Laurence C. Smith [3,4], Åsa K. Rennermalm [5], Jonathan C. Ryan [6], Lincoln H. Pitcher[1], Glen E. Liston[7], Clément Miège[5,8], Sarah W. Cooley [6] & Dirk van As [9]

The contribution of Greenland Ice Sheet meltwater runoff to global sea-level rise is accelerating due to increased melting of its bare-ice ablation zone. There is growing evidence, however, that climate models overestimate runoff from this critical area of the ice sheet. Climate models traditionally assume that all bare-ice runoff enters the ocean, unlike porous firn, in which some meltwater is retained and/or refrozen. We used field measurements and numerical modeling to reveal that extensive retention and refreezing also occurs in bare glacier ice. We found that, from 2009 to 2018, meltwater refreezing in bare, porous glacier ice reduced runoff by an estimated 11–17 Gt a$^{-1}$ in southwest Greenland alone, equivalent to 9–15% of this sector's annual meltwater runoff simulated by climate models. This mass retention explains evidence from prior studies of runoff overestimation on bare ice by current generation climate models and may represent an overlooked buffer on projected runoff increases. Inclusion of bare-ice retention and refreezing processes in climate models therefore has immediate potential to improve forecasts of ice sheet runoff and its contribution to sea-level rise.

Greenland Ice Sheet (GrIS) mass loss raised global sea level approximately 10.8 ± 0.9 mm from 1992 to 2018, and climate models forecast an additional increase of 70–130 mm by year 2100[1]. Most of this predicted mass loss derives from increased meltwater runoff from the ice sheet's ablation zone[2–6]. Within this critically important zone, winter snowpack melts entirely each summer, exposing dark, bare glacier ice which absorbs up to three times more sunlight than bright snow[7]. Warming air temperatures and reduced summer snowfall have exposed larger areas of bare ice in recent decades[2,3,8], driving enhanced surface melting in the ablation zone[2–4]. Understanding the fate of meltwater from Greenland's growing bare-ice zone is therefore critical for accurate modeling of sea levels[9].

Climate models track energy and mass budgets on the ice sheet surface and are the primary tools available to predict future ice sheet runoff[10,11]. However, there is substantial observational evidence that these models overestimate runoff from bare ice surfaces[12–17]. In the ablation zone of Greenland's melt-intensive southwest sector, proglacial and supraglacial river discharge measurements reveal up to 67% less annual meltwater release to surrounding oceans than climate model calculations[12,14–16]. Supraglacial lakes which form on the ice sheet surface fill at slower rates than predicted by climate models[17], and direct measurements of supraglacial runoff are overestimated by 21–58% during peak summer melt conditions[13]. Ice sheet mass changes are overestimated by 21–47% relative to GRACE satellite gravity retrievals[18,19], and satellite laser altimetry measurements indicate that surface melt rates are overestimated by 14–40%[20]. Similar discrepancies arise from comparisons with point ablation stake measurements, ranging from an underestimation of 17% at one ablation-

[1]Department of Geography, University of California, Los Angeles, CA, USA. [2]Sierra Crest Analytics, Portland, OR, USA. [3]Institute at Brown for Environment and Society, Brown University, Providence, RI, USA. [4]Department of Earth, Environmental and Planetary Sciences, Brown University, Providence, RI, USA. [5]Department of Geography, Rutgers, The State University of New Jersey, New Brunswick, NJ, USA. [6]Division of Earth and Climate Sciences, Nicholas School of the Environment, Duke University, Durham, NC, USA. [7]Cooperative Institute for Research in the Atmosphere, Colorado State University, Fort Collins, CO, USA. [8]Department of Geography, University of Utah, Salt Lake City, UT, USA. [9]Geological Survey of Denmark and Greenland, Copenhagen, Denmark. ✉e-mail: matt@sierracrestanalytics.com

zone site to overestimations of 10–43% at four other sites[21,22]. These discrepancies do not appear to be explained by errors in modeled surface energy balance components, which largely match in situ meteorological observations[13,21,22].

A more plausible explanation for overestimation of ice sheet runoff by climate models may be their conceptual treatment of bare ice surfaces. Climate models traditionally treat bare ice as an impervious, high-density substrate with no capacity to retain water[11,23,24]. Accordingly, runoff produced on bare ice is instantly credited in its entirety to sea level[23], despite growing field reports of non-trivial meltwater retention on or within bare ice[12,13,25–29]. These studies suggest that bare ice surfaces are not necessarily impervious, but can behave rather like snow and firn, with some portion of generated meltwater retained either through refreezing or liquid storage in pore spaces[30,31]. Because the majority (>78%)[24] of Greenland's meltwater runoff is generated from bare ice, even small amounts of meltwater retention or refreezing would constitute a substantial (and overlooked) component of the ice sheet surface mass balance that may explain consistent reports of climate model runoff overestimation[12,13,17].

Here, we formally test this hypothesis of retention and refreezing by pairing field measurements of meltwater runoff with a numerical surface mass balance model explicitly designed to simulate spectral radiation and thermodynamic heat transfer in bare glacier ice (Supplementary Materials). We compare meltwater runoff simulated by our model with outputs from two regional climate models, a global climate reanalysis, and in situ measurements from a well-studied surface

catchment on the southwest GrIS ablation zone—including supraglacial river discharge[13], surface ablation, and physical bare ice properties[26,32]. We then demonstrate that discrepancies between modeled and observed ice sheet runoff can be attributed to retention and refreezing processes in bare ice of the southwest GrIS ablation zone, where the majority of GrIS surface mass loss originates[2,3].

## Results

### Field evidence of meltwater refreezing in bare glacier ice

In July 2016, we conducted a field experiment at the "RB" catchment in southwest Greenland (Fig. 1, see also Fig. S1). This catchment covered about 60–63 km² of ice sheet surface and an elevation range of about 1200 to 1360 meters above sea level (m a.s.l.), depending on the year[13,17,33]. Meltwater runoff generated within this catchment drained through a supraglacial river network leading to a terminal moulin—a vertical shaft where surface water enters the subglacial system (Fig. 1a). During a previous field study[13] in July 2015, we measured supraglacial river discharge at a cross-section upstream from the terminal moulin. Note that discharge refers to the volumetric flow rate, measured as the volume of water passing through a channel cross-section per unit of time, whereas runoff represents the volume of water produced over the upstream contributing area per unit of time. In this paper, 'observed runoff' denotes measured discharge (in m³/s) from upstream contributing areas, which is then converted to cumulative volume (m³, and ultimately Gt) for comparison with climate model runoff (Methods).

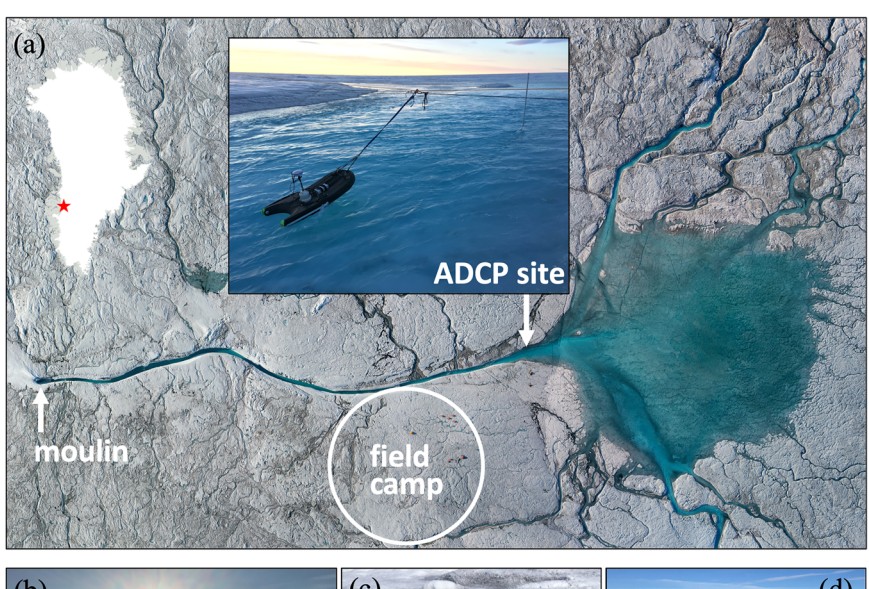

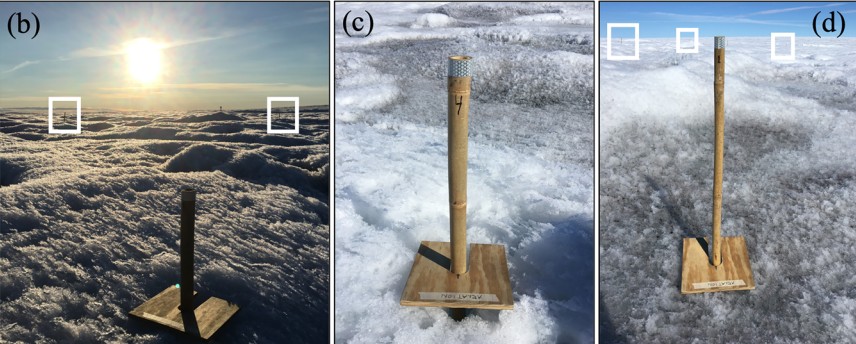

**Fig. 1 | RB catchment discharge and ablation stake measurements. a** Northward-oriented image mosaic of the RB catchment field site from aerial imagery collected during the July 2016 field experiment in southwest Greenland. White arrows mark the Acoustic Doppler Current Profiler (ADCP) discharge measurement cross-section and terminal moulin locations (see black star labeled RB in Supplementary Fig. 1). The white circle indicates the field camp and adjacent ablation stake network, with field tents visible in the upper right. The inset shows the ADCP in operation. **b**–**d** Twelve bamboo ablation stakes, used to calculate cumulative surface lowering for comparison with simulated melt rates, were distributed within an ~0.1 km² area near the field camp at safe distances from flowing water. White boxes indicate stakes visible in the background. All images were collected by the first author during the 6–13 July 2016 field experiment.

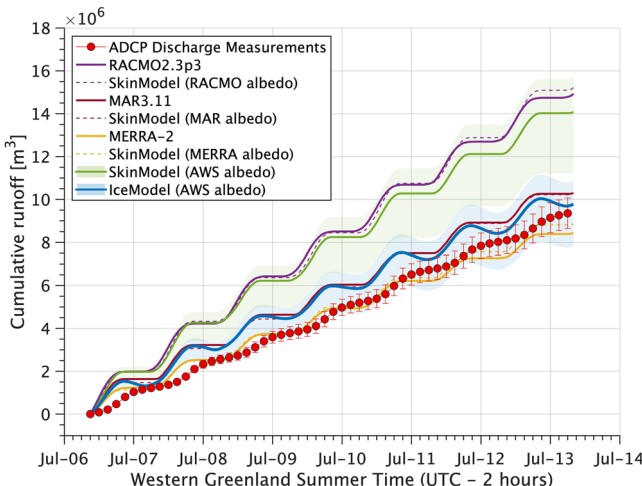

**Fig. 2 | Climate model meltwater runoff predictions compared with direct measurements on the Greenland Ice Sheet.** Cumulative values of meltwater runoff measured with an Acoustic Doppler Current Profiler (ADCP) compared with runoff predictions from three climate models and simulations from SkinModel and IceModel during the July 2016 field experiment. Dashed lines depict runoff predictions from SkinModel, our climate model surface energy balance emulator, forced with albedo outputs from each climate model. Solid lines with shaded bounds depict runoff predictions from SkinModel and from IceModel, our sub-surface energy balance model of meltwater production and refreezing, both forced with actual albedo recorded by the KAN_M automatic weather station (AWS). Solid and dashed lines indicate central runoff estimates for the 63.6 km² RB catchment area (Fig. S1, Table S1), shaded bounds depict area uncertainty (50.9–70.8 km²) (Fig. S3 depicts area uncertainty for RACMO, MAR, and MERRA−2). Error bars represent two standard deviations of ADCP discharge (Methods). For clarity, hourly discharge measurements are plotted at three-hour intervals, with major ticks and dates at 00:00 local time (UTC-2).

Our field measurements of RB catchment runoff in July 2015 revealed a substantial overestimation in predicted runoff by regional and global climate models, ranging from +21−58%[13]. To investigate this discrepancy, we revisited the site from 6 to 13 July 2016, collecting 168 consecutive hourly measurements of supraglacial river discharge[34] concurrent with three-hourly measurements of ice surface lowering[35] from a network of ablation stakes (Fig. 1b−d) we installed near the gauging site, which comprise the key field datasets of this study (Methods). Shallow ice cores obtained at the same site during July 2016 revealed a porous and meltwater-saturated weathering crust on the bare-ice surface, with an average density of just 690 kg m⁻³ within the top meter of ice[26]. Critically, we observed refrozen meltwater within this weathering crust (Figure. S2), leading to our hypothesis that the refreezing of meltwater in bare ice could explain observed discrepancies between climate model predictions and runoff observations[13,26].

## Climate models overestimate meltwater runoff from the bare-ice ablation zone

Consistent with previous studies[12–17], we find evidence that climate models overpredict observed meltwater runoff (Fig. 2). By the end of the 6–13 July 2016 field experiment, climate model runoff ranges from 7% lower (MERRA−2) to 58% higher (RACMO2.3p3) than observations (Fig. 2), similar to +21−58% we previously reported for 2015[13]. Among the climate models examined here, RACMO2.3p3 most closely reproduces observed albedo (Fig. S4) and net radiation (Figs. S5−S6), yet severely overpredicts cumulative runoff despite accurate representation of these critical surface energy balance components.

To explain these discrepancies we found between climate model predictions and runoff observations, we developed 'SkinModel', a process-based numerical model of ice sheet meltwater runoff

(Methods). SkinModel represents the ice sheet surface as an infinitely thin, hydrologically impermeable 'skin' layer of high-density (900 kg m⁻³) ice, emulating assumptions used in the current generation of climate models (Methods Eq. 1)[23]. Note that SkinModel is not a statistical emulator, but rather a simplified numerical model designed to reproduce the surface energy balance method of calculating ice sheet meltwater runoff used in climate models.

SkinModel forcings for RB catchment were provided by meteorological observations from the PROMICE/GAP KAN_M automatic weather station[36] located ~2 km from the RB catchment's northeastern boundary (67.067ºN, −48.835ºW; 1270 m. a.s.l.) (Fig. S1). To establish SkinModel's ability to effectively emulate climate model meltwater runoff predictions, we deliberately force it with surface albedo values from each climate model (rather than KAN_M albedo values), and find its predictions are virtually identical to climate model predictions (Fig. 2; dashed versus solid lines). Critically, all model forcings besides albedo are kept consistent across these emulator simulations, signifying that ice surface albedo alone can explain virtually all differences between climate model runoff predictions shown in Fig. 2.

Next, we force SkinModel with values of albedo recorded at KAN_M and find that modeled runoff is 42% higher than measured runoff (Fig. 2; green solid line). Weather station forcings thus yield runoff predictions much like RACMO2.3p3, the model with the most accurate albedo for this time and location (Fig. S4). Similar results are found using these same methods to reinterpret our earlier field measurements from 2015[13] (Figs. S6−S8). Even larger runoff overestimation is simulated if MODIS satellite albedo values[37] (Fig. S4) are used as model forcing. This reveals that among the climate models examined here, accurate representation of ice surface albedo leads to overpredicted runoff, whereas accurate predictions of runoff result from overestimated albedo.

## Runoff overestimation explained by meltwater refreezing in bare ice

We developed a second process-based numerical model of ice sheet meltwater runoff that we call IceModel[38] (Supplementary Materials). IceModel updates an earlier model of spectral radiative and thermodynamic heat transfer in glacier ice[39] with a field constraint[32] on shortwave radiation absorption by dark impurities present within Greenland's bare, ablating ice. In contrast to SkinModel, IceModel simulates an ice column with time-varying ice, air, water vapor, and liquid water contents (Methods Eq. 4), informed by observations of physical ice surface properties made during a multi-year series of field campaigns on the GrIS bare-ice ablation zone[12,13,16,26,32,33,40]. A critical feature of IceModel is that it allows sunlight to penetrate bare ice, thus allowing melt within the ice subsurface[41] rather than expending all available energy upon an infinitely thin 'skin' surface layer.

IceModel simulations reveal that, during daylight hours, penetration of shortwave radiation produces an approximately isothermal ice column nearly 1 m thick that stores latent heat in the form of liquid meltwater (Fig. S9), a phenomenon independently validated by our field observations of saturated porous ice extending at least one meter below the ice surface[26]. Lateral transport of meltwater generated within isothermal bare ice is constrained by its low horizontal hydraulic conductivity (~10⁻²−10⁻⁴ m h⁻¹)[42,43] and impermeable cold-ice lower boundary. At night, the cold polar boundary layer cools the ice surface to −5 °C on average during July at this location, as indicated by IceModel simulated surface temperatures which closely track observations (Fig. 3). These subfreezing temperatures drive refreezing of subsurface liquid meltwater at rates approaching 1 mm h⁻¹ between 02:00−04:00 local time, when the ice surface temperature drops as low as −6 °C. The runoff overestimation by SkinModel (Fig. 2) can therefore be explained by nocturnal refreezing of liquid meltwater stored within the upper decimeters of the porous bare ice matrix. Inclusion of this process enables IceModel to closely reproduce observed cumulative runoff, well within catchment boundary uncertainty (Fig. 2).

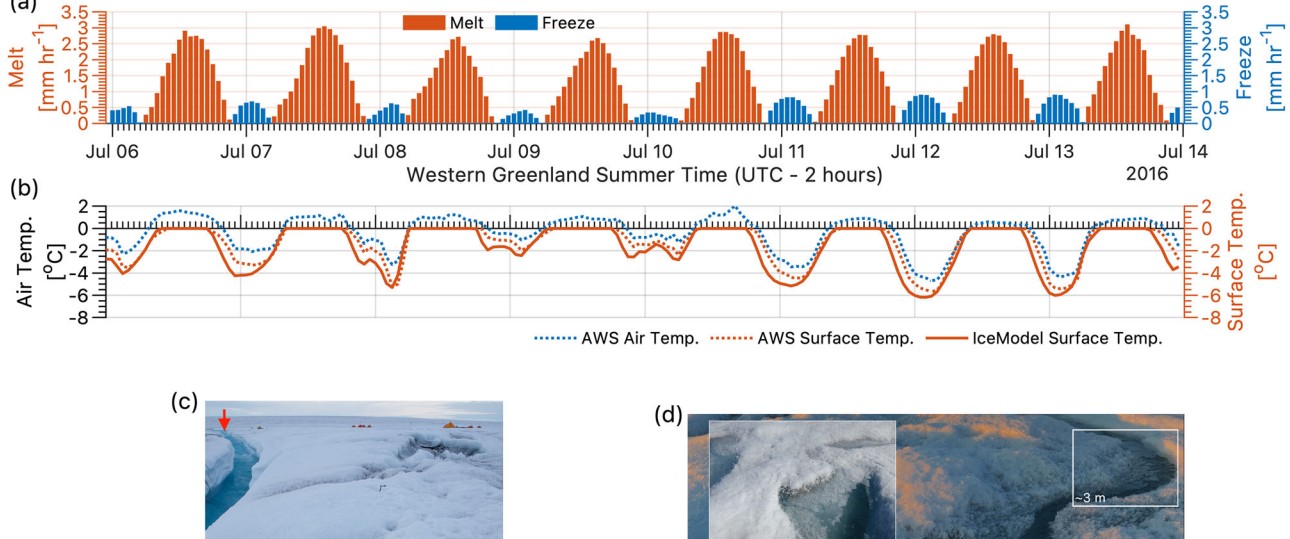

**Fig. 3 | Nocturnal refreezing of meltwater in the Greenland Ice Sheet ablation zone. a** IceModel simulations of meltwater production and refreezing, and **b** ice sheet surface temperature compared with diurnal variations in observed air and surface temperatures from the KAN_M weather station (Fig. S1) during the July 2016 field experiment. **c, d** Photographs taken during the field experiment show refrozen meltwater entrained on the weathered bare-ice surface at night and into the early morning when low sun angles and cold air kept surface temperatures

below freezing (see also Fig. S2). Inset in (**d**) shows surficial refrozen meltwater persisting to 10:00 local time on 12 July 2016 following the coldest night during the seven-day field experiment. The approximate location of the discharge gauge station is indicated by red arrow in (**c**) with field camp tents visible at right. Major ticks in (**a**) and (**b**) are at 00:00 local time (UTC-2), minor ticks are posted hourly. Photos are by the first author.

Climate model runoff over-estimations in the RB catchment (Fig. 2) are mirrored in ice surface lowering rates measured at our ablation stake network and the KAN_M automatic weather station (Fig. 4a). Unlike SkinModel and the climate model surface energy balance it emulates, IceModel predicts internal mass loss below the ice surface (Fig. 4b) caused by subsurface absorption of transmitted solar radiation. Below -0.4 m, IceModel over-predicts ice density by up to -150 kg m$^{-3}$ relative to core measurements, suggesting deeper light penetration and/or lateral meltwater delivery may increase porosity at depth more than the model allows. This bias yields a column-averaged simulated density of 770 kg m$^{-3}$–13% higher than the observed mean of 681 kg m$^{-3}$ yet still well below the canonical 900 kg m$^{-3}$ assumed in most climate models (vertical green line, Fig. 4b). Importantly, these simulations and field measurements reveal that substantial melt occurs beneath the ice surface, where meltwater is retained within porous bare ice, undetected by climate model simulations of the 'skin' energy balance, and available for nocturnal refreezing.

## Discussion

To explore the broader significance of our catchment-scale experimental conclusion that refreezing in bare ice reduces ice sheet runoff, we ran SkinModel and IceModel simulations for all bare ice areas of southwest Greenland at 5 km horizontal resolution over the decade 2009–2018 ('Methods') and compared their predictions with independent runoff measurements from two proglacial and three supraglacial catchments (Fig. 5a–b). Proglacial discharge measurements integrate supraglacial runoff routed through englacial and subglacial pathways, introducing additional uncertainty in upstream contributing area, subglacial routing, and potential groundwater fluxes (discussed further in Section S4).

Despite these complexities, our regional simulations and retrospective runoff analysis (Fig. 5) reveal a consistent climate model bias toward overestimating bare ice runoff. At LG catchment[45], which drains up to 1225 km$^2$ of the western Greenland ablation zone and spans over 1000 meters of elevation range (Fig. S1, Table S1), SkinModel forced with observed surface albedo from MODIS[37] (Fig. S10) overpredicts proglacial runoff by 21%, while IceModel underpredicts by 15% over the period 2009–2012 (Fig. 5c, Table S3). Although model bias appears less severe at LG, this could reflect uncertainty in the defined catchment area or unaccounted processes, such as subglacial melt contributions or routing delays[45]. Notably, IceModel predicts 32% less runoff than SkinModel, consistent with results from other sites where IceModel closely matches observations, such as the Akuliarusiarsuup Kuua River's northern tributary (AK4) (Fig. 5d), where a permanent streamflow gauging station provides a continuous seven-year discharge record[46]. Here, IceModel forced with MODIS albedo reproduces cumulative proglacial runoff to within 6% (Table S3), though like LG, this apparent agreement may partly reflect uncertainties in gauging station measurements, upstream contributing area, and unresolved hydrologic processes.

Although proglacial comparisons carry additional uncertainty, our results show that satellite-derived supraglacial lake volume (SLV) infilling[17]—an observational proxy for cumulative meltwater runoff—provides strong evidence of a systematic overestimation of bare ice runoff in climate models. These sites (SLV1 and SLV2; Fig. 5b) are located near this region's end-of-summer snowline elevation[2] (1520 ± 110 m a.s.l.), precisely where refreezing is expected to be most important. Notably, climate models show their largest biases at these two sites, overpredicting observed lake volume infilling by 58–81%, whereas IceModel closely tracks observations (Table S3, Section S5). Considering all runoff observations across six

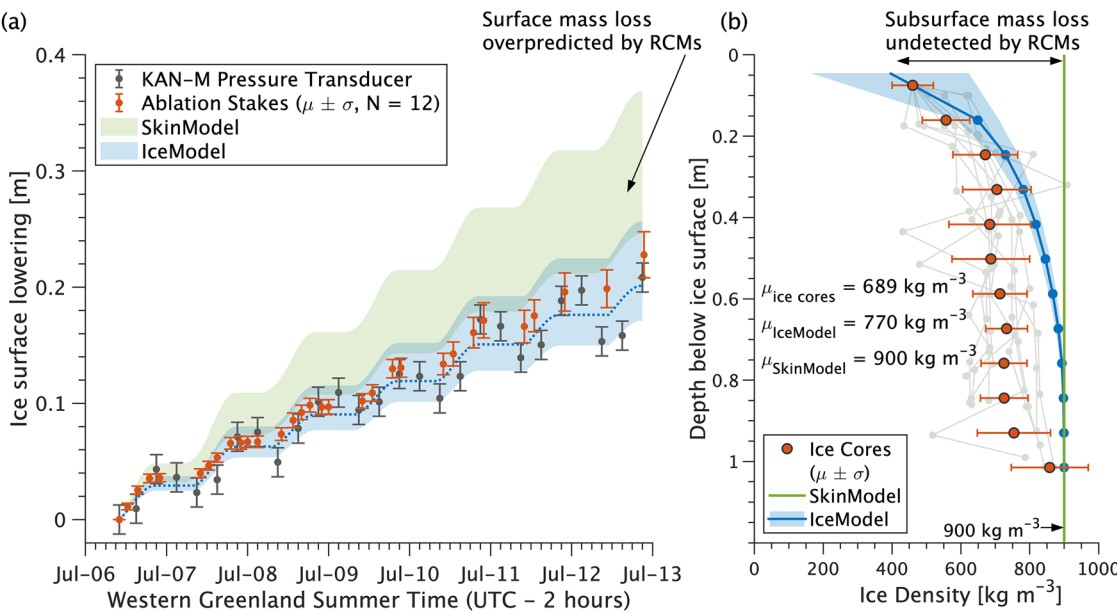

**Fig. 4 | Ice surface lowering and ice core measurements reveal subsurface ablation within bare ice on the Greenland Ice Sheet. a** Cumulative ice surface lowering recorded by the KAN_M automatic weather station (grey circles ± 2.5 cm instrument uncertainty[53]) and measured values from a network of ablation stakes in the RB experimental catchment (red circles ±1 standard deviation). Measured ablation is reproduced within measurement uncertainty by IceModel, our subsurface ice energy balance model, but is overpredicted by SkinModel, our climate model surface energy balance emulator. The green and blue envelopes depict SkinModel and IceModel net ablation (meltwater production minus refreezing) converted to ice thickness change using lower and upper assumed ice densities

(600–900 kg m$^{-3}$). The dotted blue line in (**a**) is IceModel thickness change computed directly from the modeled ice density, depicted in (**b**) by the solid blue line with shaded bounds representing ±1 standard deviation of modeled density variations during the field experiment. The modeled mean ice density within the top meter ($\mu$=770 kg m$^{-3}$) is within 12% of the empirical mean density ($\mu$=681 kg m$^{-3}$) derived from 10 ice cores sampled in the RB catchment on 11–12 July 2016[26]. Horizontal uncertainty bars in (**b**) depict mean values of ice core density ±1 standard deviation calculated from individual density profiles (grey lines in background) grouped into overlapping 15 cm depth intervals.

independent sites (Tables S1–S3), IceModel forced with MODIS albedo exhibits the lowest mean bias (-2% ± 18%), while climate models predict +9% ± 46% to + 47% ± 32% higher runoff than observations. Considering mean absolute error instead of bias yields the same qualitative conclusions.

These findings suggest that the conversion of energy (input) to mass (output) in bare ice is less efficient than currently thought, reducing the true export of meltwater runoff from the ice sheet to the ocean. Once refrozen, meltwater must thaw again before becoming runoff, an additional energy sink largely unexplored in current climate models[47], and, until now, lacking empirical verification. To demonstrate the possible magnitude of this energy sink, we estimate an annual-average runoff reduction of 11 Gt a$^{-1}$ in southwest Greenland alone due to bare ice refreezing between 15 June and 15 August over the decade 2009–2018 (Fig. S11). This is equivalent to 17% of the ~63 Gt a$^{-1}$ annual refreezing in snow and firn predicted by MAR3.11 or 9% of the ~116 Gt a$^{-1}$ total (from both ice and firn) runoff from this sector. This estimate is likely conservative because it is based on a conservative MODIS-detected bare-ice extent (Methods) and timeframe (15 June–15 August). Extending the timeframe (1 June–31 August) suggests refreezing in bare ice could reduce total runoff by up to 17 Gt a$^{-1}$ (-15% of total sector runoff, Fig. S11).

In summary, our comparison of in situ meltwater runoff measurements collected on Greenland's bare-ice surface with climate model predictions finds that climate models overestimate runoff, here by up to 58% (Fig. 2, Table S3). The two models that appear to match our in-situ runoff measurements (MERRA-2, MAR3.11) do so incidentally, via overestimated albedo (i.e., compensating errors) (Figs. 2, S4, S10). The underlying reason for this consistent model overprediction of runoff in the ablation zone[12–17] has not previously been identified, signifying a critical gap in predictive capacity concerning future ice sheet surface mass loss.

Here, we demonstrate that nocturnal refreezing of meltwater in bare-ice—a process observed on mountain glaciers[48,49] but previously unverified in Greenland—occurs at a well-studied surface catchment in southwest Greenland. Our direct supraglacial discharge measurements provide empirical evidence for this process, which may explain consistent model overestimation of bare ice runoff. While our regional analysis suggests a 9–15% runoff reduction in southwest Greenland, factors other than albedo and refreezing are likely important at larger scales. In particular, bare ice extent exerts a first-order control on runoff production[2], thus climate model runoff sensitivity likely depends not just on bare ice processes but also those which control modeled snowlines. Expanded in situ runoff measurements, coordinated model development, and controlled sensitivity tests—of climate and offline models like IceModel—are needed to fully quantify the impact of bare ice refreezing on ice sheet runoff contributions to sea level[1] and associated uncertainties[50,51].

Climate models are essential tools for estimating Greenland's meltwater runoff, and the only tool for predicting future ice sheet runoff, yet they currently overlook an important nocturnal refreezing/retention process described here. While refreezing in snow and firn may delay sea level rise by -10–17 Gt a$^{-1}$ at centennial timescales[30], our findings suggest that refreezing in bare ice may already exceed these levels. With nearly all of Greenland's meltwater now sourced from bare ice[2,24,52], this highlights an urgent need to incorporate bare-ice retention and refreezing, in addition to snow and firn processes, into future climate model projections of ice sheet mass balance.

## Methods
### Field datasets of ice sheet surface discharge, ablation rate, and ice density
Hourly supraglacial river discharges from RB catchment (Fig. S1) were measured during two field campaigns (20–23 July 2015 and 6–13 July

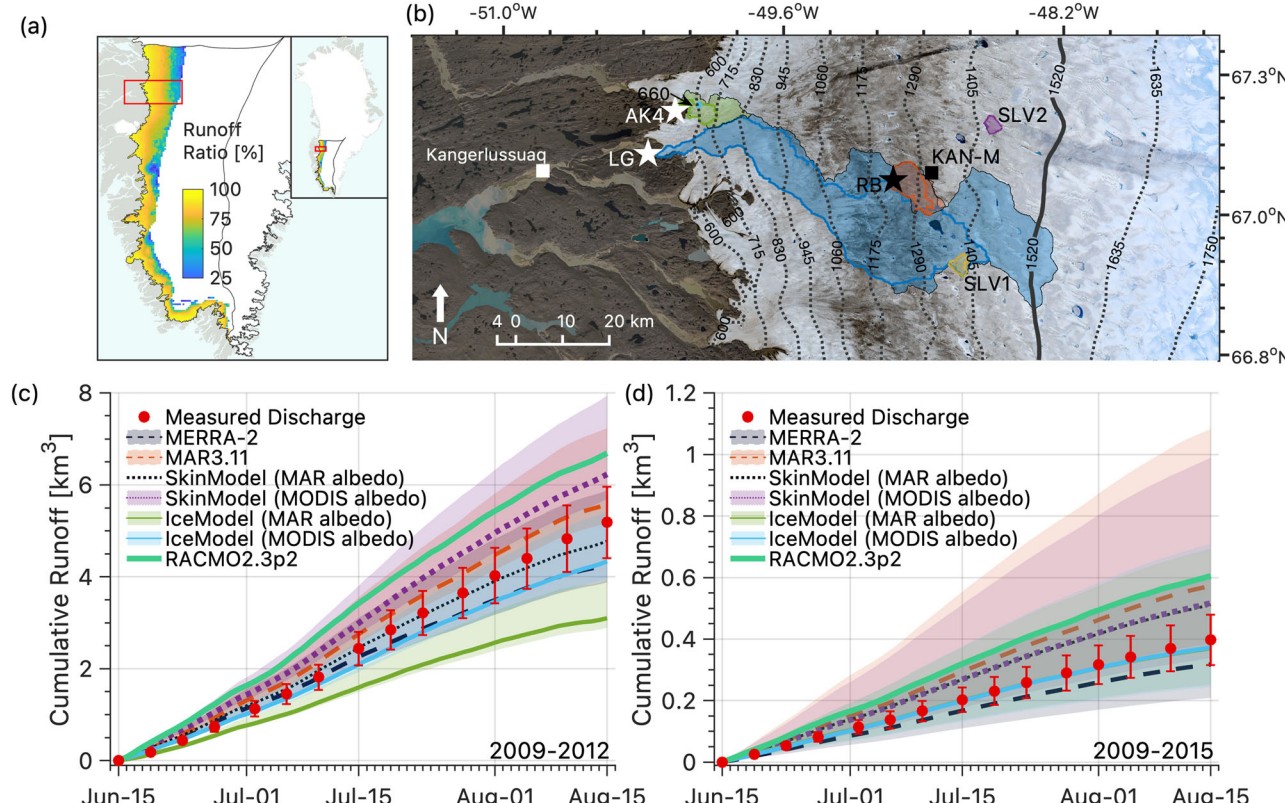

**Fig. 5 | IceModel simulations of meltwater runoff on the southwest Greenland Ice Sheet suggest an annual average runoff reduction of ~ 11–17 Gt a⁻¹ due to meltwater refreezing in bare ice. a** Runoff ratios (ratio of annual-average 15 June to 15 August cumulative bare ice runoff to meltwater production between 2009 and 2018) vary from ~100% near the ice sheet margin to ~25% near the southwest sector's annual-average snowline elevation (1520 ± 110 m a.s.l.)[2]. **b** Upper and lower catchment boundaries and discharge gauge sites for the RB, SLV1 and SLV2 supraglacial catchments, the LG and AK4 proglacial catchments, and the supraglacial 660 catchment nested inside AK4 (Tables S1–S2, Fig. S1). Background image is Landsat 8 true-color composite on 26 July 2016, with extent indicated by red box in (**a**). **c** Climate model runoff overpredicts four years of cumulative LG catchment discharge by +30% (RACMO2.3p2) and +8% (MAR3.11), similar to SkinModel (+21%),

our climate model surface energy balance numerical emulator forced with MODIS satellite ice albedo (see Fig. S10 for a comparison of MODIS albedo versus climate model albedo for the southwest sector domain and the LG catchment). IceModel forced with MODIS albedo underpredicts runoff by 15%. Vertical error bars depict discharge measurement uncertainty (±15%)[44]. **d** Climate models overpredict seven years of cumulative discharge from AK4 catchment by 53% (RACMO2.3p2) and 45% (MAR3.11), with IceModel simulations 6% lower and within measurement uncertainty[46]. Note that the two IceModel simulations yield visually indistinguishable results at AK4; see Table S3 for cumulative runoff errors to aid interpretation of panels (**c**) and (**d**). Shaded error bounds on modeled runoff depict catchment area uncertainty ('Methods').

2016)[13,34] with a SonTek RiverSurveyor M9 Acoustic Doppler Current Profiler (ADCP). The M9 was mounted on a SonTek HydroBoard II and escorted across the RB main-stem channel with a bank-operated Tyrolean system (Fig. 1). Channel geometry and flow velocity measurements were transmitted in real time to a bank-operated computer running the RiverSurveyor software. During each measurement hour, between 3 and 9 sub-hourly measurements were recorded. These raw data were converted to channel flow rate [m³ s⁻¹] in post-processing following data quality-control workflows optimized for the supraglacial environment[13,34]. The standard deviations of these sub-hourly measurements were used to estimate ± 0.6 m³ s⁻¹ measurement uncertainty. Error bars in Fig. 2 depict two standard deviations of cumulative discharge obtained by accumulating hourly discharge uncertainty.

Measurements of ice surface elevation change were provided by a network of twelve bamboo ablation stakes we installed near our field camp (67.0496° N, 49.0201° W, 1215 m a.s.l.) (Fig. 1). Stake locations were selected by generating random distance-direction pairs from a common center until a representative set of sites including bright white ice and dark impurity-laden ice were selected within an area covering ~0.1 km² (Fig. 1). Stakes were drilled 3 m deep into the ice and allowed to freeze-in for 24 hours prior to initiation of ablation stake measurements, which were then recorded at nominal 3-hour intervals

continuously from 12:00 local time on 6 July 2016 to 23:00 local time on 12 July 2016. Freeze-in was confirmed prior to each measurement to infer potential vertical displacement of these stakes due to melt-out at their base; none was observed. Prior to each measurement, a 24 × 24 cm square wooden ablation board was placed at the base of the stake and oriented to true north (see Fig. 1). This board operated as a datum from which the stake height above the ice surface was measured. Cumulative changes in stake height were converted directly to ice surface elevation change and summarized in terms of their mean and standard deviation across stakes for comparison with simulated melt rates (see Fig. 4). Field datasets of ice density from shallow ice cores, ice porosity, and ice liquid water saturation within excavated boreholes used to supplement this analysis are described in reference[26].

These direct ablation stake measurements were supplemented by hourly changes in ice surface elevation from a pressure transducer ice ablation assembly[53] installed on the PROMICE/GAP KAN_M automatic weather station[36] (67.0667° N, 48.8327° W). During July–September 2016, KAN_M station was located ~2 km from the RB catchment at an average elevation of ~1270 m a.s.l[36]. (see Fig. S1). These pressure transducer measurements have an estimated uncertainty of ± 2.5 cm liquid water equivalent[53], depicted as vertical error bars in Fig. 4a. To compare simulated melt (m liquid water equivalent) with these pressure transducer and ablation stake ice surface elevation changes, melt

was converted to ice thickness change using the conversion: $\Delta h = M(\rho_l/\rho)$, where $\Delta h$ is ice thickness change, $M$ is melt, $\rho_l = 1000$ kg m$^{-3}$ is the density of liquid water at atmospheric pressure and the triple point temperature of water (273.16 K), and $\rho$ is glacier ice density. To address known variations in near-surface $\rho$ in the study area (see Fig. 4b), uncertainty ranges on these conversions were estimated using lower and upper assumed values of $\rho$ (600–900 kg m$^{-3}$), depicted as shaded bounds in Fig. 4a. For IceModel simulations, a central estimate was obtained directly from the modeled vertical column ice density ($\rho = 770$ kg m$^{-3}$), depicted as a dotted line in Fig. 4a.

### Catchment boundaries and surface classification from satellite and airborne datasets

Catchment topography and surface classifications for RB catchment were obtained from WorldView-1 and WorldView−2 satellite images of the ice sheet surface and aerial imagery collected with an RGB sensor mounted on an uncrewed aerial vehicle[54] (Fig. S1b). These aerial images were used to reconstruct the ice sheet surface topography and to perform surface classification of snow, water, and bare ice as described in reference[13]. Surface topography was reconstructed using Agisoft PhotoScan Pro stereo-photogrammetry software. Surface classification was performed with a k-Nearest Neighbors algorithm yielding 3.1% snow cover during the 6–13 July 2016 field experiment, and 6.5% for the 20–23 July 2015 experiment[13] (Fig. S1b).

WorldView-1 and WorldView−2 satellite imagery and associated high resolution stereo-photogrammetric digital elevation models were used to delineate the RB contributing catchment area following methods in reference[13]. These methods combine traditional automated surface delineation from digital elevation with visual delineation of surface stream networks, flow direction, and channel heads visible in WorldView imagery following reference[55]. Interior channel heads (initiation points of channels that drain into the catchment) yield a minimum estimate of catchment area. Areas of internal drainage to moulins and crevasses within the catchment boundary were removed from this lower catchment area estimate (Figs. S1 and S12). Outer channel heads (initiation points of channels that drain away from the catchment) yield an upper, maximum plausible estimate of catchment area. The optimal "best guess" catchment area was delineated by tracing the digital elevation-based boundary and adjacent inner and outer channel heads in the high-resolution WorldView imagery and adding back areas that flow into the catchment and subtracting areas that flow out of the catchment. This approach yields contributing catchment area confirmed by the direction of actively flowing water tracks, an improvement upon digital elevation-based methods[56].

To complement our detailed experimental study of RB catchment, we synthesized several publicly available datasets of supraglacial and proglacial discharge measurements, for comparison with modeled meltwater runoff across the southwest sector of the ice sheet (Table S1 and Table S2). Here, *supraglacial* refers to discharge measured on the ice sheet surface, and *proglacial* to discharge measured in terrestrial rivers emerging from the ice sheet periphery which drain supraglacial runoff routed through englacial and subglacial pathways. Proglacial catchments include LG (Leverett Glacier)[44] and AK4 (Akuliarusiarsuup Kuua River's northern tributary)[46]. Supraglacial catchments include two SLV (Supraglacial Lake Volume) catchments (SLV1 and SLV2)[17], and the 660 catchment[57], a supraglacial catchment near the western ice sheet margin nested within AK4. Catchment boundaries and discharge gauge installations for these sites are depicted in Fig. S1.

Catchment boundaries for SLV1 and SLV2 were obtained using identical methods to those described above for RB catchment, provided by reference[17]. The LG and AK4 catchments boundaries were provided by the PROMICE Greenland Liquid Water Discharge dataset[58]. These delineations were produced with the GRASS GIS software single-flow direction from eight neighbors (SFD-8) drainage basin delineation

algorithm, with inputs from ArcticDEM v7 100 m gridded ice sheet surface topography and BedMachine v3[59] 150 m gridded ice thickness. The upper, middle, and lower contributing area boundaries for the LG and AK4 catchments correspond to upper, middle, and lower prescribed values of the flotation factor $k$ (1.1, 1.0, and 0.9, respectively) used in the GRASS basin delineation algorithm. These values account for the evolving subglacial water pressure and hydraulic routing throughout the summer melt season[58].

### Climate model data

Climate model outputs were provided by the Modèle Atmosphérique Régional version 3.11 (MAR3.11)[60], the polar (p) version of the Regional Atmospheric Climate Model version 2.3p3 (RACMO2.3p3)[47] and version 2.3p2 (RACMO2.3p2)[22], and the global climate model Modern-Era Retrospective analysis for Research and Applications, Version 2 (MERRA−2)[61] (Table S2). MERRA-2 data are publicly available on a global 0.5° by 0.625° latitude-longitude grid (-56 × 28 km for the grid cell intersecting RB catchment), with land ice albedo and runoff provided on a 3-hour timestep, and land surface forcings provided on a 1-hour timestep. MAR3.11 data were provided at -15 km horizontal resolution and 1 h timestep forced by European Centre for Medium-Range Weather Forecasts (ECMWF) ERA5 reanalysis[62]. RACMO2.3p3 data were provided at -11 km horizontal resolution and 3 h timestep forced with ECMWF ERA-Interim reanalysis[63]. Note that RACMO2.3p3 is the latest version of the regional climate model RACMO2 and data for the period 2009–2012 were not acquired for this study. Runoff outputs from RACMO2.3p2 were obtained from the PROMICE Greenland Liquid Water Discharge dataset[58], exclusively for comparison with LG catchment discharge during the period 2009–2012.

To address the different grid resolutions and formats of the climate models, we adopted a common grid with 5 km horizontal resolution projected onto the National Snow and Ice Data Center WGS84/NSIDC Sea Ice Polar Stereographic North (SIPSN) coordinate system (EPSG:3413). This common grid was provided with the PROMICE Greenland land-ice albedo product[37] (discussed further in Methods Section "Model simulations for individual catchments and southwest Greenland"), which was the highest-resolution gridded product used in this study. Catchment area-weighted values of climate model output were computed by intersecting the climate model horizontal grids with a bounding box surrounding each catchment boundary. The coordinates of bounded grid cell centroids and the associated gridded climate model outputs were resampled onto the common 5 km grid using Delaunay triangulation and natural neighbor interpolation. The 5 km re-gridded values were then converted to catchment area-weighted values using a conservative remapping of the overlapping area of each grid cell and the catchment polygon (Fig. S12 and Section S6). Catchment areas were computed in the National Snow and Ice Data Center Equal-Area Scalable Earth version 2.0[64] projection (Table S1).

### Comparing climate model runoff with measured discharge

The climate models evaluated in this study do not represent the processes of lateral flow, channel routing, or vertical infiltration of meltwater within bare ice. Consequently, these processes were not represented by either IceModel or SkinModel. Although the MAR model includes a routing delay designed to simulate the passage of meltwater through the ice sheet to its periphery[13], this mechanism is not applicable at supraglacial catchment scales, nor was it applied to the MAR runoff evaluated here. To compare modeled runoff with measured discharge—which inherently includes real-world catchment routing delays[65]—we focused our analysis on cumulative values [m$^3$] of runoff and discharge, obtained by summing hourly fluxes [m h$^{-1}$] multiplied by upstream contributing area [m$^2$]. This approach is necessary due to the intrinsic discrepancy between instantaneous runoff and discharge rates and aligns with our primary objective of

assessing the impact of meltwater refreezing in bare ice on total runoff from the ablation zone.

Our methodology for comparing cumulative runoff to cumulative discharge is reflected in our primary evaluation metric—the percent difference (over/underestimation) between cumulative values of modeled runoff and measured discharge (Table S3). To account for diurnal variations in runoff and routing delays, we calculated these percent differences not at a single final timestep, but as averages over the last 24 h of each observation period, which vary in length from approximately three days (RB catchment in 2015) to seven years (AK4 catchment) (Table S2). This averaging mitigates the chance alignment (or misalignment) of modeled runoff and measured discharge when assessing their agreement on the final timestep, while retaining a focus on the total meltwater runoff over the observation period.

Operational climate models including MAR3.11, MERRA-2, and RACMO2.3p2 currently do not include mechanisms for subsurface (internal) radiative heating in bare glacier ice and its consequent effects on meltwater runoff and refreezing. This simplification of bare ice persists despite emerging interest from the climate modeling community, as evidenced by a recent early inclusion of refreezing in RACMO2.3p3[47], which has yet to undergo empirical validation. Although we emphasize our independent implementation of these processes in IceModel and its empirical validation, for this study, we obtained two configurations of the RACMO2.3p3 model output: a standard version incorporating internal energy absorption in bare ice, provided for the period 2012–2018, and a control version without these processes referred to as RACMO2.3p3-WIE[47] (Without Internal Energy), provided exclusively for the period 2012–2015. Runoff outputs from these two configurations are compared with field observations in Section S5 and Table S3 (see also Table S2).

## SkinModel and IceModel description

SkinModel and IceModel both solve the zero-dimensional surface energy balance[66]:

$$\chi Q_{si}(1-\alpha) + Q_{li} - \varepsilon\sigma T_{sfc}^4 + Q_h + Q_e + Q_c = Q_m \tag{1}$$

where $\chi$ [unitless] allocates the incoming shortwave solar radiation flux, $Q_{si}$ [W m$^{-2}$], into a 'skin' surface component and a subsurface component, $\alpha$ [unitless] is ice surface albedo, $Q_{li}$ [W m$^{-2}$] is incoming longwave radiation flux, $\varepsilon\sigma T_{sfc}^4$ [W m$^{-2}$] is longwave radiation flux emitted by the ice surface, $\varepsilon = 0.98$ [unitless] is ice surface emissivity, $\sigma$ [W m$^{-2}$ K$^{-4}$] is the Stefan-Boltzmann constant, $T_{sfc}$ [K] is ice surface temperature, $Q_h$ [W m$^{-2}$] is sensible heat flux, $Q_e$ [W m$^{-2}$] is latent heat flux, $Q_c$ [W m$^{-2}$] is conductive heat flux at the surface, and $Q_m$ [W m$^{-2}$] is energy available for meltwater production. For SkinModel simulations, we set $\chi = 1$ following a traditional surface energy balance approach[39], meaning all shortwave solar radiation is absorbed at the surface. For IceModel simulations, $\chi$ was computed directly from the fraction of solar radiation absorbed in the top layer of the one-dimensional ice column, updated at each timestep with a two-stream radiative transfer model described in the Supplementary Text (Section S3). Monin-Obukhov similarity theory was used to obtain $Q_h$ and $Q_e$ as described in ref. 39.

SkinModel calculates $Q_c$ by solving the one-dimensional heat transfer equation:

$$\rho_i c_i \frac{\partial T_i}{\partial t} = -\frac{\partial}{\partial z}\left[k_i \frac{\partial T_i}{\partial z}\right] \tag{2}$$

where $\rho_i = 900$ [kg m$^{-3}$] is a reference value for glacier ice density, $c_i = 2093$ [J kg$^{-1}$ K$^{-1}$] is specific heat capacity of ice at 273.16 K and constant pressure, $T_i$ [K] is ice temperature, $t$ [s] is time, $z$ [m] is the vertical coordinate (positive down), and the glacier ice thermal

conductivity, $k_i$ [W m$^{-1}$ K$^{-1}$], was modeled as a function of ice temperature[67]:

$$k_i = 9.828\exp\left[-0.0057\,T_i\right]. \tag{3}$$

IceModel calculates $Q_c$ and subsurface meltwater production/refreezing by solving the one-dimensional thermal and spectral radiative heat transfer equation[39,68]:

$$\frac{\partial H}{\partial t} = -\frac{\partial}{\partial z}\left[k_e \frac{\partial T_i}{\partial z}\right] - \frac{\partial q}{\partial z} \tag{4}$$

where $H$ [J m$^{-3}$] is enthalpy per unit volume of the ice, liquid water, and water vapor mixture, $k_e$ [W m$^{-1}$ K$^{-1}$] is the effective thermal conductivity of the ice, liquid water, and water vapor mixture, and $q$ [W m$^{-2}$] is the subsurface net solar radiative flux at depth $z$. The model updates an earlier version[39] with field-measured values for the spectral absorption coefficient of glacier ice[69] and a conservative enthalpy formulation[70,71]. Detailed descriptions of IceModel and its numerical implementation are available in the Supplementary Text (Section S3).

Equation 2 and Eq. 4 were solved on a uniform mesh with 4 cm node spacing to a depth of 20 m, using a fully implicit time integration and a 15 min timestep. The upper boundary condition, $T_{sfc}$, was estimated by solving Eq. 1 by Newton-Raphson iteration[39]. At the lower boundary, a zero heat flux condition was assumed. Hourly values for near-surface air temperature, wind speed, relative humidity, air pressure, $\alpha$, and $Q_{si}$ and $Q_{li}$ to solve Eq. 1 were provided by the KAN_M automatic weather station for RB simulations and by MAR3.11 for regional simulations with daily MODIS (Moderate Resolution Imaging Spectroradiometer) albedo $\alpha$ from the PROMICE Greenland land-ice albedo product[37]. These hourly forcings and daily albedo values were linearly interpolated to the 15 min model timestep.

The net solar radiative flux divergence, $\partial q/\partial z$ [W m$^{-3}$], was evaluated with a two-stream radiative transfer model with 118 spectral bands[39,72] (Supplementary Text Section S3). A key input to this model is the spectral solar radiation extinction coefficient, $k_\lambda$, which controls the vertical distribution of absorbed solar radiation within subsurface ice layers. To account for the melt-enhancing effect of dark impurities present within glacier ice, values for $k_\lambda$ were obtained from direct measurements of spectral flux extinction within glacier ice in Greenland's western ablation zone[32]. Absorption by impurities was further constrained by KAN_M automatic weather station and MODIS albedo values prescribed at the upper boundary of the two-stream model.

## Model simulations for individual catchments and southwest Greenland

IceModel and SkinModel were solved on a 15 min timestep for the experimental periods 6–13 July 2016 and 20–23 July 2015 for the RB catchment, with model inputs from the KAN_M automatic weather station[36], supplemented by catchment area-weighted climate model output (e.g., climate model albedo for SkinModel emulator simulations). These simulations informed our initial catchment-scale experimental analysis depicted in Figs. 2–4.

Regional-scale simulations for Greenland's southwest sector depicted in Fig. 5 were solved on a 15 min timestep for the decade 2009–2018 by running IceModel and SkinModel at 5 km horizontal grid spacing, with each grid cell representing an independent simulation of one-dimensional vertical heat transfer (i.e., without horizontal heat transfer). The model domain was Ice Sheet Mass Balance Intercomparison Experiment (IMBIE)[1] GrIS southwest sector, which covers ~12% of the ice sheet surface area but accounts for ~34% of modeled surface runoff[24]. Model forcings for these simulations were provided by MAR3.11 on an hourly timestep (linearly interpolated to 15 min) for the

decade 2009–2018 (RACMO data were provided on a three-hour time-step for 2012–2018; relative humidity, wind speed, and surface pressure were not provided). Quality-controlled daily ice sheet surface albedo from MODIS was provided by the PROMICE Greenland land-ice albedo product[37] on a 5 km horizontal grid, which was used as a common grid for both regional-scale simulations and for resampling climate model outputs (see also Methods Section "Climate model data").

Two sets of IceModel and SkinModel simulations were performed for the southwest sector domain: one in which MAR3.11 data were used exclusively as model forcing, and a second in which MODIS albedo was used in place of MAR3.11 albedo (see Fig. 5 and Table S3). These simulations were designed to isolate the critical role of albedo in modulating runoff predictions, and to mirror our catchment-scale simulations in which climate model albedo values were used in place of weather station albedo values (see Fig. 2). Note that IceModel forced with observed surface albedo from MODIS serves as our benchmark simulation.

Southwest sector IceModel and SkinModel simulations were restricted to grid cells contained within the Greenland Ice Mapping Project (GIMP)[73] ice mask with MODIS albedo between 0.29 and 0.55 on at least 75% of days during July and August from 2000 to 2018, following refs. 2,50. This procedure was used to isolate bare-ice grid cells from firn-covered grid cells and terrestrial land. The 75% bare-ice frequency threshold is more conservative than the 50% threshold used in the GrIS SMB Intercomparison (GrSMBMIP)[50], and yielded a 61,975 km$^2$ bare-ice surface area for the southwest sector, used as a common mask for computing regional-scale bare-ice runoff and refreezing estimates (e.g., Figs. 5a and S11).

IceModel and SkinModel runoff estimates for RB, LG, AK4, 660, SLV1, and SLV2 catchments depicted in Fig. 5 and Section S5: Supplementary Runoff Comparison were computed from these gridded regional-scale simulations using the catchment area-weighted conservative remapping described in Methods Section "Climate model data". Regional-scale runoff and refreezing estimates reported in the Main text (e.g., Fig. 5) and in Fig. S11 were estimated for two time periods to illustrate a range of plausible values of bare ice refreezing and associated runoff reduction: a "lower" (15 June to 15 August) conservative proxy for seasonal bare-ice exposure duration, and an "upper" (1 June to 31 August) traditional one[2]. IceModel runoff and refreezing estimates for these two time periods were obtained directly from these regional-scale gridded simulations, restricted to grid cells within the common bare-ice mask.

## Data availability

Supraglacial river discharges and catchment boundaries for RB catchment[74] are archived at the Arctic Data Center https://doi.org/10.18739/A22F7JS1B. Ice density data[75] are archived at https://doi.org/10.1594/PANGAEA.886748. Ice ablation data[35] are archived at https://zenodo.org/records/11270233. Supraglacial river discharges and catchment boundaries for 660 catchment[57] are archived at the Arctic Data Center https://doi.org/10.18739/A2XW47X5F. Proglacial river discharges for Akuliarusiarsuup Kuua[46] are archived at https://doi.org/10.1594/PANGAEA.876357. Proglacial river discharges for Leverett Glacier[76] are available from https://doi.org/10.5285/17c400f1-ed6d-4d5a-a51f-aad9ee61ce3d. Automatic weather station data[36] provided by the Programme for Monitoring of the Greenland Ice Sheet (PRO-MICE) and the Greenland Analogue Project (GAP), Greenland land-ice albedo[37], and RACMO2.3p2 runoff and LG and AK4 catchment boundaries from the PROMICE Greenland Liquid Water Discharge dataset[58] are available from https://www.promice.org/download-data/. Bare ice classifications from MODIS imagery are available on request from Dr. Jonathan C. Ryan. MAR3.11 data were provided on request from Dr. Xavier Fettweis. RACMO2.3p3 data were provided on request from Dr. Willem Jan van de Berg. MERRA-2 data are archived at the NASA Goddard Earth Sciences and Data Information Services Center https://disc.gsfc.nasa.gov/datasets?project=MERRA-2.

## Code availability

IceModel (v1.0.0) is archived at https://zenodo.org/records/11539330.

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

## Acknowledgements
This project was funded by the NASA Cryospheric Science Program grants NNX14AH93G (L.C.S.), 80NSSC19K0942 (L.C.S.), and 80NSSC25K7960 (L.C.S.), and the NASA Earth and Space Sciences Fellowship Program grant 80NSSC17K0374 (M.G.C.). Brandon T. Overstreet contributed to field work, measurement design, and field safety protocols. Charles Kershner, Sasha Z. Leidman, Rohi Muthyala and Kang Yang contributed to field work. Polar Field Services and Kangerlussuaq International Science Support provided field logistics.

## Author contributions
M.G.C. wrote the numerical models, performed the modeling, wrote the manuscript and prepared the figures. M.G.C. and L.C.S. designed the model experiment and analysis. L.C.S., A.K.R., L.H P., M.G.C., J.C.R., C.M., and S.W.C. carried out the field experiment.G.E.L. assisted with numerical model development. J.C.R. performed satellite classification of bare ice. D.V.A. provided SEB model output. M.G.C., L.C.S., A.K.R., J.C.R., L.H.P., G.E.L., C.M., S.W.C., and D.V.A. edited the manuscript and contributed to research questions.

## Competing interests
The authors have no competing interests.
