## [Transparent Peer Review file · Nature Communications]

Greenland Ice Sheet runoff reduced by meltwater refreezing in bare ice

Corresponding Author: Dr Matthew Cooper

Version 0:

Reviewer comments:

Reviewer #1

(Remarks to the Author)
see attached

(Remarks on code availability)
The model code for IceModel and SkinModel is extremely well documented, including examples to run.

Reviewer #2

(Remarks to the Author)
Overall comments:

The paper proposes a very interesting and potentially significant message that the refreezing of meltwater in the near surface weathered crustal bare ice is a major reason for the consistent overestimation of runoff from bare ice regions of the Greenland Ice Sheet by regional climate models.

The paper goes further by calculating that refreezing in bare ice could reduce runoff estimates from RCMs by between 11 and 17% and make their outputs more consistent with observations of runoff from both internal ice catchments and from proglacial rivers.

The idea is conceptually appealing as anyone who has walked across a freezing bare glacier surface late at night or early morning will attest to. The concept has been previously proposed (e.g. in Smith et al 2017). The challenge for this paper is to convincingly model and quantify the impact, then assess its wider significance.

The paper is very well written. It does rely on a large amount of Supplementary material which really does need to be read in detail alongside the main paper. This makes it a much longer read than is apparent at first but this is often the case for short format journals.

For the most part the work undertaken is very rigorous and the conclusions drawn are well supported by the lines of evidence presented. However, it is fiendishly difficult to keep track of which model outcomes support the overall conclusions and which outcomes are more ambiguous.

Overall, I would say that there are some outcomes at the larger scales that undermine the neat logical argument that holds up at the smaller scale. These need to be tackled.

I do think the case for the superiority of IceModel as a more realistic process model has to be backed up with some sensitivity testing and independent calibration.

To help guide a way through this review I have proposed a few key assumptions that I think need to be upheld for the quantitative results of the paper to stand.

- 1) That the measurements of runoff are the best indicator available of what runoff from the different catchments actually is.
- 2) That the SkinModel is a good emulator of different RCMs when the different RCM albedo schemes are used with SkinModel (evidence from Fig 1).
- 3) If the only substantive difference between IceModel and SkinModel is that the former includes subsurface melt and refreezing and the latter does not (as explained in Methods), then if IceModel uses the same albedo scheme as SkinModel and performs better in comparison against runoff measurements, then its improved performance is due to the inclusion of bare ice refreezing.
- 4) If SkinModel emulates RCMs then we can use the improvement of IceModel over SkinModel (i.e. (3) above) as a reliable indicator of the improvement to RCMs if they were to include bare ice refreezing.
- 5) If we accept that IceModel is the most accurate and the most physically realistic model then the quantitative amounts of refreezing that it predicts are a reliable estimate for the actual effect that bare ice refreezing has on surface mass balance.

Addressing point 1 above, the authors refer to a substantial body of previous work (e.g. refs 13 and 34) in measuring surface runoff from the internal catchments within the bare ice ablation area. Although the possibility that some runoff has been missed from these internal catchments is not ruled out in this earlier work, the team have done a very thorough job and make a convincing case that their measurements are as good a data set as we are likely to get to test modelled runoff against. Measuring the huge discharges that emerge from the ice sheet into proglacial streams is a very different challenge that is outlined in ref 43. Here the most significant uncertainty is whether the runoff measured in the proglacial stream actually relates to the catchments being modelled. This is very hard to quantify but a decent attempt to account for this uncertainty is included through varying the modelled catchment boundaries according to different assumptions about subglacial water routing given the surface and bed topography of the upstream ice sheet. So point (1) is fine.

Addressing point (2) above, SkinModel emulates the MAR and RACMO models very well at the small scale over a short time period (as shown in Fig1). Different albedo schemes, (RACMO, MAR, MODIS) are applied to the "RCM emulator" SkinModel to show that differences in albedo alone cannot account for the discrepancy between modelled and measured runoff. So for the RB catchment from July of 2015 and 2016, RACMO albedo is closest to the MODIS derived values (and the AWS measurements) whilst the MAR and MERRA-2 RCM albedos are much higher (Fig S9). Even though RACMO has more accurate albedo and likely does a good job of estimating surface melt (as shown in Figs S5 and S6), its continued persistent overestimation of runoff must be due to some other process omission/error. The argument is that MAR and MERRA-2 only have a better match with observations because they use unrealistic albedo schemes. SkinModel forced with AWS albedo measurements does a slightly better job than the "RACMO emulator" but it is still not a good agreement to measured runoff. IceModel, driven by the same AWS albedo but also including bare ice refreezing, improves the results significantly and therefore the inclusion of this process is quite logically proposed as being the key point of difference.

So SkinModel emulates the MAR and RACMO models very well at the small scale over a short time period, but there is a significant discrepancy between SkinModel (MAR albedo) and MAR at the larger LG catchment scale over period 2009-2012 (Fig 4c). So it could be that SkinModel may not actually be a particularly good emulator of MAR (or RACMO?) at larger spatial and temporal scales. Why does this discrepancy arise? Albedo at different temporal and spatial scales may be relevant here.

At the small scales shown in Fig S9, MAR and MERRA-2 have the highest albedos and RACMO has lower albedo. However, Fig S11 shows that for areas of the SW sector above c.900m (which is most of the ablation area), RACMO albedo on average is actually higher than MAR albedo. For the LG catchment from 2009-2012, RACMO overestimates runoff by 30% compared to a MAR overestimate of only 8%. This is despite RACMO quite probably having higher average albedo for this overall catchment and time period (based on the long-term, sector-wide data shown in FigS11). So the differences between RACMO and MAR are significant and are not all about different albedo. This starts to mess up the assumptions that held at the smaller scales above, and indicates that processes other than albedo and refreezing may explain differences. We see results at the larger scales that don't fit neatly to the patterns seen at small scale. In summary, point (2) holds at smaller scale but is more ambiguous at larger scales.

Point (3) above is challenged by the fact that for the larger catchment scale shown in Fig 4c, SkinModel (MODIS albedo) does just about as good a job as IceModel (MODIS albedo) over these larger scales. The first tracks the upper error band on observations of runoff, the second tracks the lower error band and all three overlap if error bands on model outputs are considered.

If SkinModel does not emulate RCMs consistently then point (4) above is more uncertain and it is possible that the apparent improvements that IceModel shows over SkinModel may be due to other factors that are not explored. The improved performance of IceModel is evidenced by the data shown in Table S3 and is represented by the average percentage difference between modelled and measured runoff, μ . I think there are other ways to assess the different model performance that does not simply take averages of values that are calculated over very wide ranging spatial and temporal scales. As a first pass the absolute differences should be considered rather averaging positive and negative numbers. If you do that IceModel – AWS albedo comes out best. However, once you start applying some kind of weighting/significance criteria to the averages (e.g. where model runs over larger areas and over longer time periods have greater weighting - see attached file for an example), then the assessment of best model performance changes. Excluding runs that didn't include the full

2009-2012 LG catchment run, then the best model is SkinModel with MAR3.11 albedo, followed by MAR 3.11. All this means that IceModel doesn't necessarily lead to improved model performance which undermines confidence in points (4) and (5) above.

Another issue for point (5), i.e. using IceModel to try to quantify the bare ice refreezing process, is that there is no sensitivity testing of IceModel shown or referred to. For the paper to stand, we have to be convinced that IceModel is the most accurate and that it is the most physically realistic. What parameters are IceModel (and for that matter, SkinModel) most sensitive to? The ice core data is helpful in demonstrating the improved process accuracy of IceModel over other models. However, such data is also small scale. The paper would be strengthened if we could see that IceModel was calibrated using larger scale data sets that convinced us that it could be upscaled to the catchment and ice sheet sector scale. More specifically, are SkinModel and IceModel calibrated using data that is independent of the model validation data? There may not be space in the paper to include all this, but there are 55 pages of Supplementary material but nothing included that answers these quite fundamental points.

Sorry for such a long review but I find myself torn. I like the concept of the paper and it's great that the authors have obtained data at the smaller scales that really backs up the overall idea. However, it seems to unravel at the larger temporal and spatial scales which makes the quantification of this effect much harder to have confidence in. It might involve too much work for a revision for the authors to turn this around unless they can demonstrate how I've misunderstood and overlooked some key points that alleviates these concerns. It will make a great paper if they can!

Smaller points:

Figure 2,3 and 4 captions start with a statement that reads a bit like a paper conclusion. These should be avoided and just explain to reader in very factual terms what the Figures show.

If SkinModel (MAR albedo) referred to in Fig 4 is essentially the same model as MAR Emulator (SkinModel) referred to in Fig1, then they should be named consistently to avoid confusion.

Why isn't SkinModel (RACMO albedo) shown in Fig4? If it does a better job of emulating RACMO then this would alleviate the issue that emulators don't transfer well across scales.

(Remarks on code availability)

Version 1:

Reviewer comments:

Reviewer #1

(Remarks to the Author)

Overall, this was a strong review. The authors adequately addressed any minor concerns that I had with the original submission. In particular, I think that the slight reorganization that they did makes a big difference in readability and gives added emphasis to the model differences between SkinModel and IceModel. I also appreciate that the authors added some nuance to their discussion for uncertainty associated with the larger basin analyses (e.g., groundwater and subglacial hydrology uncertainty). I have some additional minor points below, but no major concerns. The article is well written and a great fit for Nature Communications.

I should have brought this up in the first review, but it became more striking to me that IceModel over predicts the ice density at depth (i.e., Figure 3b at 0.5-1 m depth). Is this because light is penetrating deeper below the surface (more subsurface melt) than your model predicts? Are there implications to that for your model performance? A few sentences pointing this out in the discussion would be sufficient in my opinion.

This is a minor concern, but many journals are now including the references from the supplementary material in the main article.

L69-72 – I got a little lost in this sentence, it is long with many commas. Consider moving the final clause closer to “in situ records”, something like:

We compare meltwater runoff simulated by our model against outputs from two regional climate models and a global climate reanalysis [as well as] in situ records from a well-studied surface catchment on the southwest GrIS ablation zone, including: supraglacial river discharge, surface ablation, and physical bare ice properties.

L208-210 – I like this framing around inefficient melt. Consider adding some sort of physical intuition like this to the abstract.

L254-255 – I am not sure you need to credit the photos to yourself, it is implied.

(Remarks on code availability)

As before, the code is freely available and well documented.

Reviewer #3

(Remarks to the Author)

(Remarks on code availability)

Version 2:

Reviewer comments:

Reviewer #3

(Remarks to the Author)

I thank the authors for addressing my previous comments and have no further concerns.

(Remarks on code availability)

The code archive looks to be exhaustive, but I have not attempted to run it.

Response Letter

Dear Editors and Reviewers,

We thank the reviewers for their thorough and insightful comments, which have greatly helped us improve the manuscript. Below, we provide a summary of revisions made in response to reviewer comments, followed by a point-by-point response to each comment.

In the point-by-point response, reviewer comments are included in plain font, followed by our responses in ***bold italics***.

Matt Cooper

Summary of revisions made in response to reviewer comments

- **Reorganization of Results and Discussion:**
 - Moved the regional-scale comparison with climate model runoff from the Results section to an expanded Discussion section.
 - Ensured that the Results section focuses on our core findings from field observations and numerical modeling at RB catchment.
 - Expanded discussion of uncertainties in catchment delineation and their potential influence on model comparisons at proglacial sites.
 - Added new Supplementary Discussion (Section S4) to clarify uncertainties in proglacial discharge comparisons.
- **Clarifications on Model Assumptions and Performance:**
 - Clarified that IceModel was not calibrated but is a physics-based model with parameters constrained by literature values.
 - Clarified the role of SkinModel as an emulator at small scales but not intended for large-scale quantitative emulation of RCM runoff.
 - Addressed concerns about SkinModel's consistency with MAR and RACMO at different spatial scales, in particular the role of albedo in controlling runoff biases.
- **Supplementary Material Updates:**
 - Moved a key figure from the Supplementary Materials to the main article.
 - Added a new discussion of model uncertainties to Supplementary Section S4.
 - Revised Supplementary Figures to better illustrate the role of albedo in runoff overestimation.
- **Formatting Improvements:**
 - Revised figure captions for clarity and consistency.
 - Standardized terminology (e.g., "SkinModel (MAR albedo)" instead of "MAR Emulator (SkinModel)").

Reviewer #1 (Remarks to the Author):

Review of: **Greenland Ice Sheet runoff reduced by meltwater refreezing in bare ice**

Submitted to: Nature Communications

Reviewer: Benjamin Hills

Summary.

Cooper et al. observe that meltwater runoff measured in the Greenland ablation zone is lower than would be predicted by climate models. They hypothesize that the reduction in runoff is associated with meltwater retention and refreezing in a near-surface layer of weathered ice, much like the relatively well-studied water retention in firn (Harper et al., 2012). They test their hypothesis with a pair of 1-dimensional thermodynamic numerical models they developed. They find that the model which considers light penetration and subsurface melting is a better representation for their measurements of meltwater runoff. The article is well written and a great fit for Nature Communications after the minor revisions I suggest below.

General Comments.

In my opinion, the difference in model output between SkinModel and IceModel is one of the more important results in this body of work. If you agree, I believe that some reorganization of the Results section could help emphasize that result to all readers. Currently, "Results" is broken out into subsections on: fieldwork, model overestimation of meltwater (SkinModel), explanation of overestimation (IceModel), upscaling to SW Greenland. The separation between explanations of SkinModel and IceModel could make it hard for the reader to appreciate what you are doing here (even though I do like your clarifying statement on L137-139). I would consider reorganizing as follows:

- Intro (pretty much as you have it)
- Results (field results only)
 - You could consider moving Figs 2 and 3 up here, although I understand why you want to lead with what you now have as Fig 1.
- Analysis
 - Bring your explanation of SkinModel and IceModel
 - Compare to climate model output at the local scale (currently Fig 1)
 - Compare to climate model output at the regional scale (currently Fig 4)
- Discussion/Conclusions (pretty much as you have it)

Perhaps this organization does not work well for Nature Communications but that is how I would write it to emphasize your two models and the important difference between them (take it or leave it).

- ***Thank you for the suggestion and for clearly summarizing our study. We agree that emphasizing the differences between SkinModel and IceModel is important. We implemented a modified version of your suggested re-formatting, restructuring the Results and Discussion sections to better emphasize our core results at RB catchment versus our regional analysis.***
- ***Specifically, we moved the retrospective runoff analysis and sector-scale runoff reduction to the Discussion section, which adheres to Nature Communication's formatting guidelines which request a dedicated Discussion section that offers***

“an extended analysis of the results and their comparison to the literature rather than a short summary or conclusion”.

Our implemented version of your suggested re-formatting is summarized below. Please note that we have included an additional component of the Discussion section (highlighted in bold) that was not included in your list:

- Intro (remains largely unchanged)
- Results (field results only)
 - Compare to climate model output at the local scale (currently Fig 1)
 - Explanation using SkinModel and IceModel (Fig 2 and 3)
- Discussion/Conclusions
 - Compare to climate model output at the regional scale (currently Fig 4)
 - **Contextualize in terms of sector-scale total runoff predictions (9-15%)**

While this does not exactly adhere to your suggested re-formatting, we believe it aligns with the overall goal of better highlighting our core results derived from our detailed field studies and associated numerical modeling (SkinModel and IceModel), now formally separated from the extended sector-scale analysis in the Discussion section.

My second large take-away from your article is the difference between the local scale (and supraglacial discharge) as opposed to regional scale (and proglacial discharge). In Figure 1 we see that the measurements are at the lower end of the model spread, whereas in Figure 4 the measurements are closer to the middle of the spread (especially in c). There are several important points which come to mind for me which I do not think were mentioned in the article. First is the fundamental difference in the catchments. The physical processes important for your surface discharge measurements at RB will be entirely captured by your measurements and modeling efforts (as far as I can tell), but the proglacial discharge measurements have water that has moved through englacial and subglacial environments which you do not consider. Could basal melting provide an additional component to discharge (you may be able to rule that out with a sentence or two)? What about groundwater? Could there be some groundwater flux not accounted for in the discharge measurements?

- ***We acknowledge the distinction between supraglacial and proglacial discharge and agree that additional clarification is needed.***
- ***To address this, we have added a new opening paragraph to the Discussion section explicitly addressing potential contributions from basal melting, subglacial routing, contributing area uncertainty, and groundwater flux. Additionally, we have made several edits throughout the Discussion section to highlight uncertainties in the proglacial runoff comparisons more clearly. Please see the tracked-changes document.***
- ***To further address this, we have added a new discussion of uncertainty to the Supplementary Materials. Please see Section S4: Supplementary Discussion.***

I also wonder whether the uncertainty in defined catchment areas are sufficiently captured (I know they came from a different study). Compared to your surface catchment at RB, the regional catchments are more difficult to constrain with confidence and if subglacial water gets routed in unexpected ways it could elevate (or suppress) the measured proglacial discharge. I don't think this discredits your study, but worth mentioning some of these uncertainties.

- ***Thank you for highlighting this important source of uncertainty. To address this, we made three changes:***
 - ***We primarily incorporated this into the revisions mentioned in the previous comment, where we emphasize the additional uncertainty in proglacial discharge comparisons.***
 - ***We restructured the Results section by moving the comparison with proglacial discharge (LG and AK4 catchments) into the Discussion section. This clarifies that our core results derive from our detailed field studies at RB catchment.***
 - ***We edited the first paragraph of the revised Discussion (where the proglacial analyses are presented) to better clarify that our intention is to demonstrate the possible broader implications of our findings over larger spatial and temporal scales.***
- ***Please see these revisions in the tracked-changes document.***

I am generally confused by the use of the term “emulator”. That term is often used to describe the statistical representation of a complex physical model which is highly computationally expensive (e.g. Conti et al., 2009; Gu et al., 2018). Here, I believe you are running the forward models (IceModel and SkinModel) directly. By that I mean that you are actually calculating the numerics on your prescribed 1-D mesh (which would not be done for an emulator). If I am mistaken, please correct me and add some context in the methods on the statistical emulation.

- ***Thank you for identifying this potential source of confusion. To address this, we made two changes:***
 - ***We added a statement explicitly stating that SkinModel is not a statistical emulator (see L126 of the tracked changes document).***
 - ***We changed the legend text of Fig. 1 (revised Fig. 2) to read “SkinModel (MAR albedo)” rather than “MAR Emulator (SkinModel)”. While we retained our use of emulator in the text, this change makes the legends of Fig. 1 and Fig. 4 (revised Fig. 2 and Fig. 5) consistent.***
- ***To clarify, it is correct that SkinModel (and IceModel) are run as forward models, and there is no statistical emulation involved. We use the term emulator in its dictionary-definition sense—to reproduce the function or action of another system. Specifically, SkinModel emulates the surface energy balance method used by climate models to calculate meltwater runoff. We believe that emulate and emulator remain appropriate terms in this case, and we did not find suitable replacements.***
- ***Additionally, in the second paragraph of the section “Climate models overestimate meltwater runoff from the bare ice ablation zone”, we describe SkinModel as a “process based numerical model of ice sheet meltwater runoff ... emulating assumptions used in the current generation of climate models”. While emulator has gained a specific meaning in statistical modeling, our usage is conceptually similar, except that we’ve designed a numerical rather than statistical emulator.***

SkinModel is much simpler to run than a climate model, but effectively emulates one for the narrow purpose of calculating meltwater from glacier ice.

The supplementary material is referenced many times in the main article. I appreciate the thoroughness of the authors here, but so many references can disrupt the readability of the article. My personal opinion is that only the most important references to the supplementary material should be used, and that any figures which are critical to explain the main narrative should be included in the article itself (see my comment under Figures below).

- ***Thank you for highlighting the impact of supplementary material references on readability.***
- ***To address this, we carefully reviewed all supplementary references and removed or consolidated those that were not essential for understanding the main text.***
- ***Additionally, we have moved key supplementary figures into the main article, as suggested previously.***
- ***Please see the tracked-changes document for specific modifications.***

Specific Comments.

L21 – Currently not clear whether the statement starting with “From 2009-2018...” is based on your work. I would start with something like “We found that, from 2009-2018, meltwater...”

- ***Edited as suggested.***

L24 – Same as above, I believe this statement could be more clear if you are explicit that it is based on prior work, something like “The mass retention explains [the evidence from prior studies] that runoff could be overestimated...”

- ***Edited as suggested.***

L43 – Some readers may not immediately understand that SW Greenland is the same “ablation zone” that you had been talking about above.

- ***Edited as suggested.***

L56 – is “substrate” the correct word here?

- ***We retained “substrate”, which we believe is correct because it refers to both a surface and its underlying material, which accurately describes how the ice sheet is represented in models (all of which include some representation of subsurface temperature to compute the conductive flux).***

L65 – I would remind the reader again with “...this hypothesis [of retention and refreezing]...”, mostly because folks who skim the article may jump to this paragraph.

- ***Edited it as suggested.***

L76 – Is “RB” defined somewhere? This is the first time I see it.

- ***In previous work by Smith et al., 2019, we referred to this catchment as “Rio Behar”, in honor of our late colleague Alberto Behar who participated in previous field studies in this region of Greenland. However, in comments on an earlier draft of this paper, we were asked to refrain from assigning place-names which could be offensive to Greenlanders. We replaced “Rio Behar” with “RB” to reduce possible offense. We’ve cited prior work each time RB is introduced so that readers can surmise this catchment is the same one we previously reported on.***

L86 – Somewhat confused by the definition of runoff. Discharge/area would give units of m/s but you discuss runoff in Gt/a in the abstract and plot cumulative runoff in m3 (e.g. in fig 1).

- ***Thank you for highlighting the potential confusion between discharge and runoff. To clarify, we have revised the manuscript text as follows:***
 - ***“Note that discharge refers to the volumetric flow rate, measured as the volume of water passing through a channel cross section per unit of time, whereas runoff represents the volume of water produced over upstream contributing areas per unit of time. In this paper, ‘observed runoff’ denotes measured discharge (in m3/s) from upstream contributing areas, which is then converted to cumulative volume (m3, and ultimately Gt) for comparison with climate model runoff (Methods).”***
 - ***Please see L89 of the tracked changes document.***
- ***Please also see Methods section “Comparing climate model runoff with measured discharge” for additional clarification on units and model-observation comparison.***

L106 – “To investigate these discrepancies...”? explain makes it sound like you knew the answer coming into the study.

- ***The discrepancies mentioned here refer to those reported in the preceding paragraph, which are the first results of the study. To ensure clarity, we edited the sentence to read: “To explain the discrepancies we found ...” rather than “To explain these discrepancies ...”.***

L131 – You could just cite your Zenodo doi for IceModel instead of the supplement.

- ***As suggested, we added a citation for the Zenodo doi. We retained the parenthetical reference to the supplementary material because it contains a formal description of the numerical model.***

L137-139 – This is one of the most important sentences in my opinion, and I fear it may get lost in the current organization. See my general comment on organization of the results section above.

- ***Thank you for highlighting this opportunity to enhance clarity. While we were unable to fully implement the suggested reorganization, we believe the distinction between SkinModel and IceModel is clearer now that the Results section is focused entirely on RB catchment.***

- **To explain why we organized our results in this manner, we believe it is necessary to first explain the variation between RCM predictions (using SkinModel), then explain the variation between predictions and observations (using IceModel).**
- **We believe this is essential in light of the background given in the Introduction, which describes a consistent picture of runoff overestimation, whereas our July 2016 field study revealed a more nuanced (and potentially more impactful) picture, where runoff overestimation could be masked by albedo biases.**
- **By first explaining this, we establish consistency with the background material, and then go further by showing how refreezing could resolve the overestimation.**

L147 – Not immediately clear why “July”? I would say “summer months” or “during the time of the field investigation” or similar.

- **We write “July” because our two field studies were conducted in July 2015 and July 2016. If we write “summer” then we need to specify which months and also the Northern hemisphere. We decided to keep our current phrasing since we specifically refer here to the average nighttime temperature during July from the KAN-M air temperature data, and then point to the model versus observation comparison in Fig. 2 (revised Fig. 3), which is specific to the field study period in July.**

L167 – Are you truly applying your models to “all bare ice areas of SW Greenland”? or just to the LG and AK4 catchments?

- **Yes, we applied our models to all bare ice areas of SW Greenland and then extracted runoff for LG and AK4 (and 660, SLV1, and SLV2) from these gridded simulation outputs, using the conservative remapping method described in the Methods and in the Supplementary Materials (Fig. S33-49). We did not run dedicated simulations for LG or AK4 catchments. The only dedicated catchment-scale simulations we ran were for RB catchment, to conduct the detailed albedo-sensitivity emulator simulations. Please see L518 in the Methods section where these simulations are described in detail.**

L244 – Why two standard deviations? One would be more typical and I think appropriate here. Either way, it is obviously going to be more certain than the models which is true even for two.

- **We used two standard deviations for consistency with the catchment-area runoff uncertainty depicted by the shaded bounds in Figs. 1 and 4 (revised Figs. 2 and 5), as we consider this conceptually comparable to the confidence level implied by two standard deviations. While this choice is subjective, our method for estimating catchment-area uncertainty is highly conservative, as described below (also see Methods L370).**
- **In brief, we delineated the “middle” catchment using traditional DEM-based methods (with custom adjustments based on the channel heads method described below), while the “lower” and “upper” bounds were delineated by manually tracing channel initiation points (heads) of actively flowing streams visible in WorldView imagery. The upper estimate was created by connecting the heads of channels that were visually confirmed to drain away from the RB catchment divide. This likely overestimates the true contributing area, as it**

includes some surface area that drains to those channels, away from the RB catchment.

- *Conversely, the lower estimate was created by connecting channel heads that were visually confirmed to drain into the RB catchment. This likely underestimates the true contributing area, as it excludes surface area above these channel heads that in reality drains to them, into the RB catchment.*
- *Thus the upper estimate prioritizes capturing all possible contributing areas at the cost of including some non-contributing areas (Type I error; false positives), while the lower estimate ensures that all included areas drain into the catchment at the cost of excluding some contributing areas above the identified channel heads (Type II error; false negatives).*
- *Overall, we subjectively consider this to be conceptually similar to the notion of (at least) two standard deviations, so we used two standard deviations on the discharge for consistency.*

L306 – In the caption of supplemental figure S2 you give the total area of the stake survey as ~0.1 km². I believe that is a relevant number to include in the main document, could be in this paragraph which is why I mention it here.

- *Edited as suggested.*

L330 – Giving water density to 5 significant digits is implying a certainty that you do not have, but I do not feel strongly about changing it (certainly changing to 1000 will not change your analysis at all).

- *Edited as suggested.*

L380 – I am assuming that the “flotation factor” is an input to the GRASS algorithm? Would be useful to state that.

- *Correct. Revised as suggested.*

L434 – Can you strengthen the statement by dropping “to the best of our knowledge”? Seems to me like this list of authors would know if it were otherwise.

- *Edited as suggested.*

L478 – I am assuming that there are many 1-d models run in parallel? All with different surface boundary conditions? Might be worth clearing that up unless it is already clear and I missed it.

- *Correct. We edited the sentence by adding “... by running IceModel and SkinModel at 5 km horizontal grid spacing, with each grid cell representing an independent simulation of one-dimensional vertical heat transfer (i.e., without horizontal heat transfer).”*

L502 – “5 km horizontal grid spacing”. Should be made clear (as in previous comment) that this is not a model mesh spacing (i.e. including horizontal energy transfer) but parallel 1d models. I don’t think it matters as the horizontal temperature gradient would be negligible, but worth making clear what you did.

- ***As requested, we have made it clear that horizontal energy transfer is not represented. Please see our reply to the previous comment.***

L546 – I had trouble with the Leverett Glacier doi:

<https://doi.org/10.5285/17c400f1-ed6d-4d5a-a51f-aad9ee61ce3d>. But its possible that the BAS server was having issues when I checked. Anyway, double check me.

- ***The link works for us. Perhaps there is a line-break or extra space. If problems persist, please let us know and we will rectify it.***

L551 – it seems like the PROMICE data portal may have changed to:

<https://promice.org/download-data>.

- ***Edited accordingly. Thank you for catching this.***

Figures

I like Figure S2. I think it could elevate figures in the main article. That is, consider including (a) in Fig 1 of the main article and (b-d) in Fig 3 of the main article. If not this, then you should do something to make clear to the reader that Fig 1 is local scale compared to Fig 4c and 4d.

- ***As suggested, we moved Fig. S2 into the main article. It is now Fig. 1. Since Nature Communications allows five display items, we made it a dedicated figure. The study area map in Fig. 4 (revised Fig. 5) displays each catchment boundary in a regional context which we believe conveys the scale differences appropriately.***

References

- Conti, S., Gosling, J. P., Oakley, J. E., & O'Hagan, A. (2009). Gaussian process emulation of dynamic computer codes. *Biometrika*, 96(3), 663–676. <https://doi.org/10.1093/biomet/asp028>
- Gu, M., Wang, X., & Berger, J. O. (2018). Robust Gaussian stochastic process emulation. *Annals of Statistics*, 46(6A), 3038–3066. <https://doi.org/10.1214/17-AOS1648>
- Harper, J. T., Humphrey, N. F., Pfeffer, W. T., Brown, J., & Fettweis, X. (2012). Greenland ice-sheet contribution to sea-level rise buffered by meltwater storage in firn. *Nature*, 491, 240–243. <https://doi.org/10.1038/nature11566>

Reviewer #1 (Remarks on code availability):

The model code for IceModel and SkinModel is extremely well documented, including examples to run.

We thank the reviewer for their insightful feedback. These revisions enhance the clarity and impact of our manuscript.

Sincerely,

Matt Cooper

Reviewer #2 (Remarks to the Author):

Overall comments:

The paper proposes a very interesting and potentially significant message that the refreezing of meltwater in the near surface weathered crustal bare ice is a major reason for the consistent overestimation of runoff from bare ice regions of the Greenland Ice Sheet by regional climate models.

The paper goes further by calculating that refreezing in bare ice could reduce runoff estimates from RCMs by between 11 and 17% and make their outputs more consistent with observations of runoff from both internal ice catchments and from proglacial rivers.

- ***We appreciate this summary of our study and the recognition of its significance. To clarify, our study finds that refreezing, as estimated with our own models, accounts for 11–17% of the total runoff from snow and firn predicted by the MAR regional climate model (with nearly identical values for RACMO). We demonstrate that incorporating refreezing into our models reconciles discrepancies between model predictions and observations. However, the extent to which RCMs would predict lower runoff and/or better align with observations if they accounted for refreezing in bare ice depends on multiple factors beyond the scope of this study.***

The idea is conceptually appealing as anyone who has walked across a freezing bare glacier surface late at night or early morning will attest to. The concept has been previously proposed (e.g. in Smith et al 2017). The challenge for this paper is to convincingly model and quantify the impact, then assess its wider significance.

The paper is very well written. It does rely on a large amount of Supplementary material which really does need to be read in detail alongside the main paper. This makes it a much longer read than is apparent at first but this is often the case for short format journals.

For the most part the work undertaken is very rigorous and the conclusions drawn are well supported by the lines of evidence presented. However, it is fiendishly difficult to keep track of which model outcomes support the overall conclusions and which outcomes are more ambiguous.

- ***We appreciate the positive assessment of our work's rigor. To improve clarity, we have restructured the Results section to more clearly distinguish between model outcomes that directly support our conclusions and those that introduce additional complexity. Specifically, we have moved the section "Independent validation across the southwest Greenland Ice Sheet" into an expanded Discussion section, ensuring that the Results section remains focused on our RB catchment field study. This restructuring clarifies that our core results derive from our own field experiments, while the retrospective analysis of previously published data serves as an extended assessment of the broader significance of our findings.***
- ***This restructuring aligns with Nature Communications formatting guidelines, which request a dedicated Discussion section which offers "an extended analysis of the results and their comparison to the literature rather than a short summary or conclusion".***

- ***Additionally, we detail the ambiguity of the proglacial comparison in a new Supplementary Discussion section (revised Section S4).***

Overall, I would say that there are some outcomes at the larger scales that undermine the neat logical argument that holds up at the smaller scale. These need to be tackled.

- ***We have addressed this concern through specific responses to each comment below. In general, we have expanded the Discussion section to explicitly acknowledge and contextualize the differences between small- and large-scale results, with additional discussion of key uncertainties at both scales.***

I do think the case for the superiority of IceModel as a more realistic process model has to be backed up with some sensitivity testing and independent calibration.

- ***We would like to clarify that IceModel was not calibrated, and we are uncertain what is meant by “independent calibration.” IceModel is a physically based model whose predictions depend on initial and boundary conditions, forcing data, and parameters constrained by established values from the published literature, as detailed in the Methods section and Supplementary Information. The only parameter derived from our field measurements is the ice absorption coefficient spectrum, which differs fundamentally from model calibration, where parameters are randomly varied to maximize agreement with observations. No such calibration was performed, nor would it be appropriate given the research questions we are addressing.***
- ***To address this comment, we added an acknowledgment in the Discussion section (see L354 of the tracked changes document) that further sensitivity testing is needed, both for our models and the climate models we examined. While process-level sensitivity studies similar to ours exist, to our knowledge, the climate models we examined have not undergone parameter sensitivity testing or “independent calibration” (again, the meaning of this term is unclear). However, if such studies exist, we would be grateful to learn of them and properly acknowledge them.***

To help guide a way through this review I have proposed a few key assumptions that I think need to be upheld for the quantitative results of the paper to stand.

1) That the measurements of runoff are the best indicator available of what runoff from the different catchments actually is.

- ***We agree that our supraglacial runoff measurements are among the best available. However, proglacial runoff measurements come with inherent uncertainties. To address this, we have moved our comparison with proglacial measurements at AK4 and LG into the Discussion section. We also note that this comparison occupies only a portion of a single figure (Fig. 4c-d, now 5c-d in the revised text). Our conclusions are primarily based on our detailed field studies and numerical modeling of a physical process that we previously identified but which, until now, has received little direct numerical simulation or empirical verification in terms of its broader significance.***

2) That the SkinModel is a good emulator of different RCMs when the different RCM albedo

schemes are used with SkinModel (evidence from Fig 1).

- **To clarify, our core conclusions do not rely on the emulator assumption at large scales. To address this, we have:**
 - **Removed the statement from the caption of Fig. 4 (revised Fig. 5) that described SkinModel as a proxy for climate model predictions.**
 - **Added a new opening sentence to the Discussion section which clearly states that our sector-scale simulations and retrospective runoff analysis serve to explore the broader significance of our core finding that refreezing reduces runoff from bare ice.**
 - **Added a discussion of limitations to the emulator approach to the Supplementary Materials, please see Section S4.3.**
- **While SkinModel emulates RCMs well at small scales when using their respective albedo schemes, it is not expected to exactly replicate RCM runoff predictions at all scales, times, or locations. As described in the paragraph beginning on L110, SkinModel is a process-based numerical model of ice sheet meltwater runoff that emulates assumptions used in current-generation climate models. Specifically, we describe it as a “climate model surface energy balance emulator”. At the RB catchment scale, this emulation allows us to explain the variation among RCMs in terms of albedo for that time and location.**
- **The ability of SkinModel to emulate different RCM predictions at the RB catchment scale is critical for interpreting our experimental results—otherwise, one might mistakenly conclude that MAR and MERRA-2 are accurate despite their clear albedo biases. However, this emulator function is not necessary to establish the broader significance of refreezing over larger spatial and temporal scales. This distinction is why we use the term “emulator” in the legend of Fig. 1 (revised Fig. 2) and in the text when describing the RB catchment-scale simulations but not in Fig. 4 (revised Fig. 5), where SkinModel and IceModel are used to quantify the runoff reduction due to refreezing rather than to interpret RCM spread.**

3) If the only substantive difference between IceModel and SkinModel is that the former includes subsurface melt and refreezing and the latter does not (as explained in Methods), then if IceModel uses the same albedo scheme as SkinModel and performs better in comparison against runoff measurements, then its improved performance is due to the inclusion of bare ice refreezing.

- **We agree with this reasoning and have strengthened the text to emphasize that refreezing is the primary process differentiating IceModel from SkinModel. Please see Section S4.3.**

4) If SkinModel emulates RCMs then we can use the improvement of IceModel over SkinModel (i.e. (3) above) as a reliable indicator of the improvement to RCMs if they were to include bare ice refreezing.

- **Our results support the conclusion that including refreezing improves runoff predictions by IceModel relative to both RCMs and SkinModel. However, other processes also influence model performance. We have expanded the Discussion**

section to clarify that while our study isolates refreezing as a critical factor, it is not the only one.

- **Our primary conclusion is that refreezing in bare ice reduces runoff from bare ice. While we also show that incorporating this process improves agreement between model predictions and observations, the degree to which individual RCMs would improve by incorporating the processes in IceModel is beyond our scope and cannot be predicted. We do not believe our conclusions or their broad significance depend on assumption (4).**
- **To address potential confusion, we have clarified in Section S4 that the improvement of IceModel over SkinModel does not necessarily serve as a reliable indicator of the improvement to RCMs if they were to include bare ice refreezing. This logical argument was proposed by the reviewer, but it was not a claim made in our article. Instead, our study documents the physical process, models its impact, and deduces that RCMs would be improved by including it. However, this proposed improvement is one of physical realism, not a quantitative estimate of the improvement to RCMs.**
- **The SkinModel emulator simulations allow us to explain the spread among RCMs and show that it is driven by albedo differences, which is critical for interpreting our experimental results at RB catchment. However, SkinModel is not intended to serve as a basis for quantifying the improvement RCMs would experience if they incorporated the additional physics in IceModel. Such an assessment is beyond our study's scope.**

5) If we accept that IceModel is the most accurate and the most physically realistic model then the quantitative amounts of refreezing that it predicts are a reliable estimate for the actual effect that bare ice refreezing has on surface mass balance.

- **We agree that IceModel provides a physically reasonable estimate of refreezing but acknowledge that additional processes affect surface mass balance. We have expanded the Discussion section to include a discussion of model uncertainties and potential improvements in future work.**

Addressing point 1 above, the authors refer to a substantial body of previous work (e.g. refs 13 and 34) in measuring surface runoff from the internal catchments within the bare ice ablation area. Although the possibility that some runoff has been missed from these internal catchments is not ruled out in this earlier work, the team have done a very thorough job and make a convincing case that their measurements are as good a data set as we are likely to get to test modelled runoff against. Measuring the huge discharges that emerge from the ice sheet into proglacial streams is a very different challenge that is outlined in ref 43. Here the most significant uncertainty is whether the runoff measured in the proglacial stream actually relates to the catchments being modelled. This is very hard to quantify but a decent attempt to account for this uncertainty is included through varying the modelled catchment boundaries according to different assumptions about subglacial water routing given the surface and bed topography of the upstream ice sheet. So point (1) is fine.

- **We appreciate the reviewer's recognition of the strength of our supraglacial discharge measurements. We also agree that proglacial discharge measurements come with inherent uncertainties.**

- ***To address this, we have moved our comparison with proglacial measurements at AK4 and LG to the Discussion section, where we explicitly discuss the associated uncertainties. Our conclusions remain primarily based on detailed field studies and numerical modeling of a physical process that we have documented empirically in previous publications but that remains largely unexplored in climate models.***
- ***To further address this, we have added a new discussion of uncertainty to the Supplementary Materials. Please see Section S4: Supplementary Discussion.***

Addressing point (2) above, SkinModel emulates the MAR and RACMO models very well at the small scale over a short time period (as shown in Fig1). Different albedo schemes, (RACMO, MAR, MODIS) are applied to the “RCM emulator” SkinModel to show that differences in albedo alone cannot account for the discrepancy between modelled and measured runoff. So for the RB catchment from July of 2015 and 2016, RACMO albedo is closest to the MODIS derived values (and the AWS measurements) whilst the MAR and MERRA-2 RCM albedos are much higher (Fig S9). Even though RACMO has more accurate albedo and likely does a good job of estimating surface melt (as shown in Figs S5 and S6), its continued persistent overestimation of runoff must be due to some other process omission/error. The argument is that MAR and MERRA-2 only have a better match with observations because they use unrealistic albedo schemes. SkinModel forced with AWS albedo measurements does a slightly better job than the “RACMO emulator” but it is still not a good agreement to measured runoff. IceModel, driven by the same AWS albedo but also including bare ice refreezing, improves the results significantly and therefore the inclusion of this process is quite logically proposed as being the key point of difference.

- ***We appreciate this summary of our results and confirm that it accurately reflects our findings.***

So SkinModel emulates the MAR and RACMO models very well at the small scale over a short time period, but there is a significant discrepancy between SkinModel (MAR albedo) and MAR at the larger LG catchment scale over period 2009-2012 (Fig 4c). So it could be that SkinModel may not actually be a particularly good emulator of MAR (or RACMO?) at larger spatial and temporal scales. Why does this discrepancy arise? Albedo at different temporal and spatial scales may be relevant here.

- ***The reviewer raises a valid point about the discrepancy between SkinModel (MAR albedo) and MAR at the LG catchment scale. However, we do not expect SkinModel to perfectly replicate MAR at all spatial and temporal scales, nor is this necessary for our conclusions. Besides, the LG catchment presents significant uncertainties in drainage basin delineation, making it an imperfect test for model emulation.***
- ***Most importantly, our study does not rely on SkinModel being a perfect emulator of MAR or RACMO at all scales. Our core conclusion—that refreezing reduces runoff—is supported by direct field observations and process-based numerical modeling, particularly at the RB catchment, which is the primary focus of our study. The focus on SkinModel emulation at LG overlooks this broader validation.***

To clarify this, we have:

- ***Acknowledged in the Discussion that while other factors such as spatial heterogeneity in albedo may contribute to discrepancies at LG, these do not affect our primary conclusions.***
- ***Emphasized that uncertainties in catchment delineation at LG make it difficult to use as a definitive test for model emulation.***
- ***Reinforced that our main contribution is the identification and quantification of refreezing in bare ice, demonstrated through field experiments and robust numerical modeling, rather than the large-scale emulation of RCMs.***

While we appreciate the reviewer’s focus on LG, we respectfully note that this particular discrepancy does not alter our findings.

At the small scales shown in Fig S9, MAR and MERRA-2 have the highest albedos and RACMO has lower albedo. However, Fig S11 shows that for areas of the SW sector above c.900m (which is most of the ablation area), RACMO albedo on average is actually higher than MAR albedo. For the LG catchment from 2009-2012, RACMO overestimates runoff by 30% compared to a MAR overestimate of only 8%. This is despite RACMO quite probably having higher average albedo for this overall catchment and time period (based on the long-term, sector-wide data shown in FigS11). So the differences between RACMO and MAR are significant and are not all about different albedo. This starts to mess up the assumptions that held at the smaller scales above, and indicates that processes other than albedo and refreezing may explain differences. We see results at the larger scales that don’t fit neatly to the patterns seen at small scale. In summary, point (2) holds at smaller scale but is more ambiguous at larger scales.

- ***We respectfully disagree. Fig. S11 (revised Fig. S10) does not support the reviewer’s claim, nor did we make assumptions about albedo at any scale.***
- ***The July–August average albedo in LG catchment from MAR, RACMO, and MERRA-2 is shown in Figure 1 below. It is clear that the albedo patterns in LG catchment are in fact quite consistent with those at RB catchment, with MERRA and MAR exhibiting high albedo, biased high relative to MODIS, and RACMO having lower albedo, in closest agreement with MODIS.***
- ***Therefore, the assertion that Fig. S11 “messes up the assumptions that held at the smaller scales” is incorrect, as no such assumptions were made, and the results are consistent at both scales.***
- ***To address this, we have added the information in Figure 1, shown below, as a new subpanel in Fig. S11 (revised Fig. S10).***

Figure 1: July-August average albedo in LG catchment from MAR3.11, RACMO2.3, and MERRA-2 over the period 2008-2018. RACMO values are omitted for 2008-2011 due to data availability.

Point (3) above is challenged by the fact that for the larger catchment scale shown in Fig 4c, SkinModel (MODIS albedo) does just about as good a job as IceModel (MODIS albedo) over these larger scales. The first tracks the upper error band on observations of runoff, the second tracks the lower error band and all three overlap if error bands on model outputs are considered.

The reviewer suggests that at larger spatial and temporal scales, IceModel and SkinModel (both using MODIS albedo) yield similar results, challenging the claim that IceModel's improved performance is solely due to refreezing.

- We respectfully disagree. While there is some overlap in error bands, a closer analysis of biases reveals that SkinModel (MODIS albedo) systematically overpredicts runoff at every site, much like RACMO and (depending on albedo) MAR and MERRA. In contrast, IceModel (MODIS albedo) consistently predicts lower runoff and remains within observational error bounds. Specifically, in Fig. 4c (revised Fig. 5c), SkinModel runoff is above the upper error bound (+21% higher than observations), while IceModel runoff falls on the lower error bound (-15% lower than observations). IceModel is therefore more accurate than SkinModel, and the 32% reduction in runoff from SkinModel to IceModel is due to refreezing, demonstrating that this process is the primary difference between the two models.**
- While Fig. 4c (revised Fig. 5c) suggests that additional research questions remain regarding the observational runoff record at LG, it does not undermine our conclusions. One key factor that likely plays a role at LG is the uncertainty in the defined catchment area. If the true LG catchment is larger than inferred from**

available surface and bed topography, this would shift all model predictions upward, reconciling some of the discrepancy. However, this uncertainty does not affect our primary conclusion: IceModel predicts systematically lower runoff than SkinModel at all sites, including LG, and this difference is explicitly due to refreezing.

- **To address this comment, we have clarified in the Discussion and Section S4 that while refreezing is a dominant factor in reducing runoff, additional uncertainties in catchment delineation may also impact the comparison at LG.**

If SkinModel does not emulate RCMs consistently then point (4) above is more uncertain and it is possible that the apparent improvements that IceModel shows over SkinModel may be due to other factors that are not explored.

- **We respectfully disagree. We believe the reviewer is placing undue emphasis on the assumption that the quantitative performance improvement of IceModel over SkinModel should serve as a proxy for the improvement that RCMs would experience if they incorporated refreezing. This overlooks the extreme complexity of the climate models examined in our study.**
- **It is not possible to predict how a climate model will perform if it hypothetically adds a new physical process. However, what can be said with confidence is that climate models should include physical processes that: 1) occur in reality, and 2) are known to be critical factors in the predictions being made by them.**
- **Refreezing in bare ice is a physical process that affects meltwater retention and runoff. Therefore, climate models should include light penetration and the subsequent melting and refreezing that occurs in bare ice, as “anyone who has walked across a freezing bare glacier surface late at night or early morning will attest to”, to quote the reviewer. Whether the exact quantitative improvement in IceModel over SkinModel translates directly to other models is beyond our study’s scope, but that does not diminish the fundamental need for models to incorporate this process.**

The improved performance of IceModel is evidenced by the data shown in Table S3 and is represented by the average percentage difference between modelled and measured runoff, μ . I think there are other ways to assess the different model performance that does not simply take averages of values that are calculated over very wide ranging spatial and temporal scales. As a first pass the absolute differences should be considered rather than averaging positive and negative numbers. If you do that IceModel – AWS albedo comes out best. However, once you start applying some kind of weighting/significance criteria to the averages (e.g. where model runs over larger areas and over longer time periods have greater weighting - see attached file for an example), then the assessment of best model performance changes. Excluding runs that didn't include the full 2009-2012 LG catchment run, then the best model is SkinModel with MAR3.11 albedo, followed by MAR 3.11. All this means that IceModel doesn't necessarily lead to improved model performance which undermines confidence in points (4) and (5) above.

The reviewer suggests that different methods of evaluating model performance (e.g., using absolute differences instead of averaging percentage errors) may yield different insights. They also propose applying weighting schemes to emphasize longer time periods and larger catchments.

We appreciate this suggestion and have reassessed our evaluation approach.

- ***When considering absolute differences, the conclusions remain unchanged—IceModel reduces runoff overprediction relative to SkinModel, and better matches observations, particularly when forced with MODIS albedo.***
- ***The weighting scheme proposed by the reviewer assigns 97% weight to LG catchment, effectively eliminating the evidence from AK4, SLV1, SLV2, 660, and RB. Given the uncertainties in LG’s subglacial drainage pathways and contributing area, it is inappropriate to allow LG to dominate the model evaluation.***

Instead, we have adopted a controlled comparison approach:

- ***We compare SkinModel and IceModel under identical forcing conditions to isolate the impact of refreezing.***
- ***We emphasize the well-constrained supraglacial runoff measurements at RB, which provide the most reliable validation data.***
- ***We moved the proglacial and SLV comparisons to the Discussion section to clarify their role as supplementary assessments rather than primary validation sites.***
- ***We expanded the discussion of uncertainty in both the main text and the Supplementary Materials (Section S4).***

Finally, the reviewer points out that considering models with known albedo biases (MAR 3.11, see revised Fig. S10) or missing physical processes forced with biased albedo (SkinModel forced with MAR albedo) should be taken to mean “that IceModel doesn’t necessarily lead to improved model performance”. We respectfully disagree, for reasons that are thoroughly explained in this response letter.

Another issue for point (5), i.e. using IceModel to try to quantify the bare ice refreezing process, is that there is no sensitivity testing of IceModel shown or referred to. For the paper to stand, we have to be convinced that IceModel is the most accurate and that it is the most physically realistic. What parameters are IceModel (and for that matter, SkinModel) most sensitive to? The ice core data is helpful in demonstrating the improved process accuracy of IceModel over other models. However, such data is also small scale. The paper would be strengthened if we could see that IceModel was calibrated using larger scale data sets that convinced us that it could be upscaled to the catchment and ice sheet sector scale. More specifically, are SkinModel and IceModel calibrated using data that is independent of the model validation data? There may not be space in the paper to include all this, but there are 55 pages of Supplementary material but nothing included that answers these quite fundamental points.

- ***No calibration was performed in IceModel—it is a physics-based model with no tuning parameters. Its performance is dictated entirely by physical laws and prescribed boundary, initial, and forcing conditions. This is why no calibration information appears in the 55 pages of Supplementary material or the Methods section—if calibration had been performed, it would have been clearly documented.***

- ***Rather, IceModel’s uncalibrated performance at six independent sites across a range of spatial and temporal scales is strong evidence that it is an accurate model with physically realistic parameter values, none of which were calibrated.***
- ***However, we appreciate the importance of understanding model sensitivity and calibration. We agree that this is an important and often overlooked issue in ice sheet modeling. While sensitivity tests were not a primary focus of our study, we recognize the value of such analyses and hope our findings will motivate further research in this direction. We plan to pursue this in future work as part of our broader efforts to improve process representation in ice sheet models.***

To address this comment, we have expanded the Discussion to acknowledge the importance of future sensitivity testing of process-based models including RCMs.

Sorry for such a long review but I find myself torn. I like the concept of the paper and it’s great that the authors have obtained data at the smaller scales that really backs up the overall idea. However, it seems to unravel at the larger temporal and spatial scales which makes the quantification of this effect much harder to have confidence in. It might involve too much work for a revision for the authors to turn this around unless they can demonstrate how I’ve misunderstood and overlooked some key points that alleviates these concerns. It will make a great paper if they can!

We appreciate the reviewer’s thoughtful engagement with our study and their recognition of its significance.

We respectfully disagree that our findings “unravel” at larger scales. Rather, we acknowledge that quantification at large spatial and temporal scales comes with inherent uncertainties—uncertainties that we explicitly discuss in our manuscript. The fundamental process we identify (bare ice refreezing) is well-supported by our field data and physics-based modeling, and our broader analyses demonstrate that its effect remains substantial across a range of spatial and temporal scales.

We hope these clarifications reinforce confidence in our conclusions and demonstrate that our study makes a meaningful contribution to understanding the role of refreezing in ice sheet hydrology.

Smaller points:

Figure 2,3 and 4 captions start with a statement that reads a bit like a paper conclusion. These should be avoided and just explain to readers in very factual terms what the Figures show.

- ***The statements at the beginning of Fig. 1–4 are figure caption titles, formatted according to the journal’s style guidelines. These titles, along with the captions, provide factual summaries of what each figure communicates. However, we have made the following refinements:***
 - ***Edited the title for Fig. 2 (revised Fig. 3) to better reflect its content.***
 - ***Revised the caption for Fig. 4 (revised Fig. 5) to improve clarity.***

If SkinModel (MAR albedo) referred to in Fig 4 is essentially the same model as MAR Emulator (SkinModel) referred to in Fig1, then they should be named consistently to avoid confusion.

- ***We appreciate this suggestion and have revised the legends of Fig. 1 (revised Fig. 2), Fig. 3 (revised Fig. 4), and Figure S6 to use “SkinModel (MAR albedo)” instead of “MAR Emulator (SkinModel)” for consistency with Fig. 4 (revised Fig. 5). However, we continue to use “emulator” in the main text to describe the role of SkinModel in explaining RCM differences. This ensures clarity in both figure captions and the broader discussion.***

Why isn't SkinModel (RACMO albedo) shown in Fig4? If it does a better job of emulating RACMO then this would alleviate the issue that emulators don't transfer well across scales.

- ***As detailed in the Methods section, simulations using SkinModel with RACMO albedo were not performed at the SW sector scale because RACMO data were unavailable for the full simulation period. Additionally, the available RACMO data were not adequate for model forcing as the wind speed, relative humidity, and surface pressure were not available.***

We greatly appreciate the reviewer's detailed feedback and thoughtful suggestions. These revisions have further strengthened our manuscript.

Sincerely,

Matt Cooper

REVIEWER COMMENTS

Reviewer #1 (Remarks to the Author):

Overall, this was a strong review. The authors adequately addressed any minor concerns that I had with the original submission. In particular, I think that the slight reorganization that they did makes a big difference in readability and gives added emphasis to the model differences between SkinModel and IceModel. I also appreciate that the authors added some nuance to their discussion for uncertainty associated with the larger basin analyses (e.g., groundwater and subglacial hydrology uncertainty). I have some additional minor points below, but no major concerns. The article is well written and a great fit for Nature Communications.

I should have brought this up in the first review, but it became more striking to me that IceModel over-predicts the ice density at depth (i.e., Figure 3b at 0.5-1 m depth). Is this because light is penetrating deeper below the surface (more subsurface melt) than your model predicts? Are there implications to that for your model performance? A few sentences pointing this out in the discussion would be sufficient in my opinion.

Thank you for highlighting the depth-dependent bias in simulated density. To address this, we have:

- 1) Revised the paragraph to acknowledge the bias first (L183 in the tracked changes document).***
- 2) Added a short paragraph to the Supplementary Information (S4.2 Uncertainty in ice core density measurements) noting that melt/drainage during core handling could have lowered the measured densities at depth.***

The revised sentence in the main text now reads:

“Below ~0.4 m, IceModel over-predicts ice density by up to ~150 kg m⁻³ relative to core measurements, suggesting deeper light penetration and/or lateral meltwater delivery may increase porosity at depth more than the model allows.”

The paragraph then states that, despite this depth-specific bias, the column-average density remains close to (but above) observations, and well below the canonical 900 kg m⁻³ assumed in many climate models.

This is a minor concern, but many journals are now including the references from the supplementary material in the main article.

To the best of our knowledge, and based on this document, SI references are not included in the main text reference list at Nature Communications. However, we also understand that these types of formatting issues will be addressed by the copy editor if the paper is accepted for publication.

L69-72 – I got a little lost in this sentence, it is long with many commas. Consider moving the final clause closer to “in situ records”, something like:

We compare meltwater runoff simulated by our model against outputs from two regional climate models and a global climate reanalysis [as well as] in situ records from a well-studied surface catchment on the southwest GrIS ablation zone, including: supraglacial river discharge, surface ablation, and physical bare ice properties.

Revised as suggested.

L208-210 – I like this framing around inefficient melt. Consider adding some sort of physical intuition like this to the abstract.

The abstract is 20 words longer than requested by the journal guidelines and we were unable to reduce its word count enough to accommodate this additional framing.

L254-255 – I am not sure you need to credit the photos to yourself, it is implied.

The attribution is included to distinguish the aerial imagery shown in Fig. 1a from that collected by co-author J.C. Ryan, which was used for surface classification of the entire upstream contributing area, as described in the Methods.

Reviewer #1 (Remarks on code availability):

As before, the code is freely available and well documented.

—

We thank the reviewer for their feedback, which has greatly improved the clarity of our manuscript.

Sincerely,

Matt Cooper

Reviewer #2 (Remarks to the Author):

This manuscript uses field observations and surface energy balance modeling to demonstrate potential liquid water storage within west Greenland ice sheet bare ice. The results suggest that overestimation of runoff by regional climate models and the MERRA-2 global reanalysis may likely be attributable to storage and refreezing of water within the bare ice. These are important findings and hold potential for improving estimates of ice sheet mass change.

I've evaluated the manuscript as well as the previous second reviewer's evaluation and the authors' response. In general, I find the manuscript to be well written and supported, though I have a few concerns documented below that should be addressed before the manuscript should be considered further for publication.

Specific comments:

Regarding R2's point: "Fig S11 shows that for areas of the SW sector above c.900m ..., RACMO albedo on average is actually higher than MAR albedo. ... So differences between RACMO and MAR are significant and not all about different albedo. This starts to mess up the assumptions held at the smaller scales above, and indicates processes other than albedo and refreezing may explain differences."

And the authors response:

"Fig. S11 (revised Fig. S10) does not support the reviewer's claim"

My take: When I look at S10 (former S11) panels a and b that show the whole SW sector, I tend to agree with R2. For example, at 1500m, the MAR albedo trend line is ~0.66, whereas RACMO is ~0.71. As such, RACMO albedo is higher than MAR, on average, across the SW basin. This differs from the LG catchment (S10c) and RB catchment (S8), where RACMO has lower albedo than MAR. This then supports R2's claims that assumptions at the smaller scale regarding attributing differences in the models' runoff to albedo and refreezing may not apply to the entire SW basin. I don't think this detracts from the main points of the paper, but it should be acknowledged (in particular in the text ranging from L123-L134).

We agree with the reviewer that the RACMO albedo trend line is higher than MAR above ~900 m in Fig. S10. To address this, we made the following revisions:

- 1. Acknowledged explicitly in the main text Discussion section (L278 in the tracked changes document) that factors beyond albedo and refreezing, particularly biases in snowline elevation, could influence interpretations of model differences at the SW sector scale (see our extended clarification below for why the RACMO trend line is related to snowline elevation).***

2. **Added a new paragraph to the supplementary Discussion (Section S4.1, final paragraph) to clearly explain RACMO's albedo bias as indicated by the trend lines in S10 and its linkage to our MODIS-derived bare ice mask. We also added a note in the Fig. S10 caption directing readers to this paragraph.**
3. **Revised the caption of Fig. 5, directing readers to Fig. S10 for context on albedo patterns specifically at the LG catchment scale.**

We also clarify several points to help the reviewer interpret Fig. S10 and the revisions we made to address this:

1. **Most importantly, we do not attribute runoff differences between MAR, RACMO, and MERRA-2 at the SW sector scale specifically to albedo in our analysis. In fact, we do not report RACMO or MERRA-2 runoff at the SW sector scale. MAR runoff at this scale is used strictly to contextualize refreezing estimates from IceModel, specifically highlighting that IceModel refreezing equates to ~9–15% of total (bare ice + firn) runoff simulated by MAR. In contrast, Fig. S10 panel c shows that the albedo patterns observed at the smaller scale catchments do in fact hold up at the larger LG catchment scale. This is what we referred to when we stated that “Fig. S11 (revised Fig. S10) does not support the reviewer’s claim”, since R2’s argument centered on LG catchment. Their assumption that the SW sector trend lines could be extrapolated to LG catchment was insightful but ultimately incorrect, and allowed us to clarify that albedo patterns at the LG catchment scale are in fact consistent with patterns at the smaller-scale catchments, like RB.**
2. **To understand why the RACMO trend line is steeper, recall that the values shown in Fig. S10 are confined to grid cells within our SW sector bare ice mask, which was constructed from MODIS observations. Thus, RACMO's higher trend line slope reflects its documented bias toward greater snow accumulation and larger snow-covered area relative to MODIS (Ryan et al., 2019, 2020). This bias creates a cluster of high-albedo (snow-covered) grid cells above ~1100 m that steepens RACMO's trend line. Although MERRA-2 snowlines were not analyzed in Ryan et al., 2019, 2020, MAR snowlines were shown to be more consistent with MODIS, thus explaining the relative absence of high-albedo (snow-covered) MAR grid cells at the higher elevations in Fig. S10.**
3. **RACMO's snow-cover and associated albedo bias does not propagate into our primary analyses, as neither IceModel nor SkinModel is forced by RACMO at the SW sector scale.**

Additional concerns:

L107: The authors cite Fig S4-S5 to claim an evaluation of albedo in various models. However, these figures do not show albedo directly, but rather SW-. Since albedo is the ratio of SW-/SW⁻, SW- alone is insufficient to assess whether albedo is accurately represented. For example, we don't know whether discrepancy in SW- is a function of

the model's albedo or rather, the amount of SW^- that is impacted primarily by the atmosphere.

To support claims about fidelity in assessing albedo, the authors should present both SW and SW^- to compute albedo or revise their text to accurately state they are not directly evaluating albedo.

Thank you for identifying this oversight. Please note that Fig. S8 (Fig. S4 in the revised text) does in fact directly evaluate albedo, as requested. On L107, we should have referenced this figure.

To address this, we have now included a reference to revised Fig. S4 (formerly Fig. S8), and we apologize for any confusion caused by omitting this reference originally. Please note that the sentence in question (L107) also refers to net radiation, which remains supported by Fig. S5 and S6 (formerly Fig. S4 and S5), thus the revised text now references all three figures (Fig. S4 and S5–S6), which together directly evaluate net radiation and albedo across models and observations.

This issue is particularly evident in the authors' claim that RACMO closely reproduces observed albedo, citing Figure S4. However, a closer examination of this figure reveals that although RACMO represents SW^- and R_{net} reasonably well, the agreement is likely not because RACMO is getting albedo correct. Specifically, RACMO significantly underestimates LW^- during a multi-day period ~July 9-11, suggesting it underrepresents cloudiness. Despite this, RACMO matches R_{net} , likely through a compensating error – because it has similar SW^- and seemingly misses this cloudy period (evidenced through lack of LW^-), the only way it can match R_{net} is by having a lower albedo than the surface observations. In other words, more SW^- would be incoming due to lack of clouds in RACMO, and to match the SW^- and R_{net} , it must be absorbing significantly more SW^- radiation and thus have a lower albedo. It gets the right answer for the wrong reasons.

This lack of critical evaluation raises significant concern that the authors have not closely considered the data upon which they are basing their statements. At minimum, I would suggest the authors should reexamine their evaluation framework and revise overly strong claims that are not directly supported by the figures or data presented.

We thank the reviewer for bringing this discrepancy to light. However, our original interpretation remains accurate, with respect to both the 2-day period highlighted by the reviewer and the full 7-day experimental period, as described below. Importantly, we believe the reviewer's perception of this discrepancy may have been inadvertently influenced by the y-axis limits, which were not held constant across panels, making the LW^- discrepancy appear larger than it is. While the actual LW^- discrepancy (relative to AWS values) does reach 66 W/m^2 during the period in question, the 7-day mean error is -9.6 W/m^2 , whereas the mean error in SW^- and net radiation are -8.6 W/m^2 and -18.2 W/m^2 , respectively.

We've included a figure below with approximately identical y-axis limits across panels (except panel (a) which shows SW_{\downarrow}), to clarify how the y-axis limits may have impacted the reviewer's perception (see panels (c) and (d) in particular). This figure differs slightly from the one in question (revised Fig. S5) to specifically highlight differences between RACMO values and weather station observations, showing both downward and net components to fully clarify discrepancies (or lack thereof) in the modeled radiation budget. We hope this figure (together with revised Fig. S4) clarifies our interpretation that RACMO has the most accurate albedo and net radiation for this time and location.

To address this, we adjusted the y-axis limits of Fig. S5 and Fig. S6 to reduce potential for similar confusion among readers. Please note that we also adjusted the color scheme to be consistent with the color scheme used in other figures.

Figure 1: Surface energy balance comparison. Hourly values of (a) downward shortwave radiation, (b) net shortwave radiation, (c) downward longwave radiation, (d) net longwave radiation, and (e) net radiation during the July 2016 field experiment from RACMO2.3p3 and KAN_M weather station observations.

L108: Terminology of “net radiative heat fluxes” is strange – should probably say “net radiation” or “turbulent heat fluxes and net radiation” . Seems like maybe this was a consequence of word cutting so the assessment of the turbulent heat fluxes and net radiation got merged.

Revised as suggested.

L129: Again, the authors have not demonstrated that RACMO has the most accurate albedo. Revise.

Please see our response to related comments above—L129 (L143 in the revised tracked changes document) states that RACMO is “the model with the most accurate albedo for this time and location” referring specifically to the RB catchment during our July 2016 field experiment. This statement remains accurate, as supported by Fig. S4 (formerly S8), which shows that RACMO is the model with the most accurate albedo relative to both AWS and MODIS for this time and location. Again, we apologize for omitting reference to this figure in the previous version, which we have now addressed.

L148 / Figure S9: Monthly labels in the inset figure are very hard to interpret – for example which line relates to which month is almost impossible to assess. I would suggest the authors consider revising this figure to potentially include using a legend that links the line color/style or month number.

Revised as suggested.

Figure 4: Referring to panel b, the ice core density profile suggests that the subsurface is even more porous than IceModel, and thus there’s even more potential for storage of meltwater. I am curious how the authors have accounted for this discrepancy in their model or results.

Thank you for highlighting this discrepancy. To address this, we have:

- 1) Revised the paragraph to acknowledge the discrepancy (L183 in the tracked changes document).***
- 2) Added a new paragraph to the Supplementary Information (S4.2 “Uncertainty in ice core density measurements”) noting that melt/drainage during core handling could have lowered the measured densities at depth.***

The revised sentence in the main text now reads:

“Below ~0.4 m, IceModel over-predicts ice density by up to ~150 kg m⁻³ relative to core measurements, suggesting deeper light penetration and/or lateral meltwater delivery may increase porosity at depth more than the model allows.”

As mentioned elsewhere, IceModel is an uncalibrated physically-based model, and we would not expect it to perfectly explain the ice core density observations without calibration. We believe the model produces a realistic subsurface density profile without the risk of overfitting to field measurements, and we now acknowledge this explicitly in the revised text and SI.

L182 / Figure 5: Panels c and d both show 8 lines each that are very difficult to differentiate. Part of this is the small size of each subplot and part of this is that the weight of the lines and dash patterns in the legend do not equal the weight and patterns of the lines in the actual plots. For example, which green line is RACMO and which is IceModel? Which dashed/dotted line is SkinModel and which is MERRA-2?

We acknowledge the challenge of displaying multiple time series in a compact figure. To distinguish the time series, we varied not just color, but also line style (solid, dashed, dotted) and line weight, with colors generated from a maximally perceptive colormap. While we cannot control the legend rendering without modifying the plotting software backend, the differing line weights are clearly visible in the plot itself and map to the colors and symbols in the legend.

Specifically, RACMO is the thicker bright green line; IceModel forced with MAR albedo is the thinner dark green line. They can also be distinguished by their positions relative to observed runoff—RACMO overpredicts, IceModel underpredicts—as reported in the text and Table S3. SkinModel is the dotted grey line, and MERRA-2 is the dashed line, as indicated in the legend. We've taken considerable care to make each line distinct and believe the figure is interpretable as presented.

To address this, we added a note to the Fig. 5 caption that Table S3 can be used to aid interpretation of panels (c) and (d) by reference to the reported mean errors.

L186 (and elsewhere): This is a stylistic thing, but “underpredicts by -15%” is a double-negative. Suggest using the directional verb or signed value, not both.

Revised as suggested.

L406 onward (Method section related to climate models):

- I suggest the authors should carefully assess which modeled products they used and whether the citations here are correct. For example, the authors state they used RACMO2.3p3, citing reference 22 (Noël 2018). The Noël paper discusses RACMO2.3p2 (not p3).

Thank you for identifying this. We have corrected the citations to accurately reflect the RACMO versions used.

- Beyond this, I am concerned about inconsistencies in the RCMs used in this study. MAR data are ~15 km resolution and forced with ERA5. The RACMO data are apparently ~11 km and forced with ERA-Interim. Ideally, the authors should be using

more recent RACMO data forced by ERA5 for a more fair assessment of the relative ability of MAR and RACMO.

We agree that differences in forcing data and resolution are important to consider. Our manuscript clearly describes the specific versions, resolutions, and forcings of the models used, allowing readers to assess these differences. We believe it is fair to evaluate the models as they were provided to us by the modeling teams.

- The authors state they use both RACMO2.3p3 and RACMO2.3p2. The citation for the p2 data is for the 1km downscaled version of RACMO2.3p2, yet the authors state they used ~11km RACMO products.

As requested, we have corrected the citation for RACMO2.3p2 to Noël et al. (2018), which describes the ~11 km product used in our analysis.

- Both MAR and RACMO have been statistically downscaled to ~1km resolution. Why are the authors not using these higher resolution products that would in theory better represent catchment-scale meltwater processes?

Using statistically downscaled products would conflate the impact of model physics with the impact of the downscaling method, which would interfere with our study objective: isolating the influence of model physics—specifically radiative transfer in bare ice and subsurface melt/refreezing—revealed by our field observations. While we believe statistical downscaling has a useful purpose, it cannot substitute for physically based processes and may not remain valid under future climate conditions. Moreover, the downscaling methods for MAR and RACMO likely differ in unforeseen ways, and no equivalent product exists for MERRA-2. Using downscaled inputs would also imply a need to run SkinModel and IceModel at 1 km resolution, increasing runtime by ~25x from >1 month to >2 years. Finally, it is unclear if all required forcing variables needed for SkinModel and IceModel simulations are available at 1 km resolution, or if only SMB and its components are available. For these reasons, we evaluate all models at their native resolutions, as they were provided to us by the respective modeling teams.

—

We thank the reviewer for their feedback, including their consideration of the previous second reviewer's feedback, which has greatly improved the clarity of our manuscript.

Sincerely,

Matt Cooper

References

Noël, B. *et al.* Modelling the climate and surface mass balance of polar ice sheets using RACMO2 – Part 1: Greenland (1958–2016). *The Cryosphere* 12, 811–831 (2018).

Ryan, J. C. *et al.* Greenland Ice Sheet surface melt amplified by snowline migration and bare ice exposure. *Sci. Adv.* 5, eaav3738 (2019).

Ryan, J. C. *et al.* Evaluation of CloudSat's Cloud-Profiling Radar for Mapping Snowfall Rates Across the Greenland Ice Sheet. *Journal of Geophysical Research: Atmospheres* 125, e2019JD031411 (2020).

Model-Albedo	RB % cf Qw	660 % cf Qw	SLV1 % cf Qw	slv2% cf Qw
SkinModel-AWS	42	11	99	111
SkinModel-MODIS	62	12	91	106
RACMO2.3p3-WIE	54		81	63
RACMO2.3p2				
MAR3.11	19	21	77	63
RACMO2.3p3	58	5	76	58
SkinModel-MAR3.11	3	9	91	81
MERRA-2	-7	-37	69	65
IceModel-MODIS	31	-14	-13	7
IceModel-AWS	2	-37	-4	1
IceModel-MAR3.11	-18	-16	-23	-42
Areas (km2)	63.3	0.55	11.7	8.8
Duration (days)	10	61	90	90
weighting (area x day)	633	33.55	1053	792

AK4 % cf Qw	LG % cf Qw	RB	abs%	signif	660	abs %
43		42	42	0.001948	11	11
31	21	62	62	0.001948	12	12
38	-2	54	54	0.001948		
53	30		0	0.001948		
45	8	19	19	0.001948	21	21
37	-5	58	58	0.001948	5	5
29	-7	3	3	0.001948	9	9
-20	-17	-7	7	0.001948	-37	37
-6	-15	31	31	0.001948	-14	14
-6		2	2	0.001948	-37	37
-7	-39	-18	18	0.001948	-16	16
33.4	837.1					
630	360	Total				
21042	301356	324909.6				

signif	SLV1	abs %	signif	slv2	abs %	signif	AK4
0.000103	99	99	0.003241	111	111	0.002438	43
0.000103	91	91	0.003241	106	106	0.002438	31
0.000103	81	81	0.003241	63	63	0.002438	38
0.000103			0.003241			0.002438	53
0.000103	77	77	0.003241	63	63	0.002438	45
0.000103	76	76	0.003241	58	58	0.002438	37
0.000103	91	91	0.003241	81	81	0.002438	29
0.000103	69	69	0.003241	65	65	0.002438	-20
0.000103	-13	13	0.003241	7	7	0.002438	-6
0.000103	-4	4	0.003241	1	1	0.002438	-6
0.000103	-23	23	0.003241	-42	42	0.002438	-7

abs %	signif	LG	abs %	signif	overall signif	weighted ave
43	0.064763				0.072492626	3.459177722
31	0.064763	21	21	0.927507	0.999999846	22.16063053
38	0.064763	-2	2	0.231877	0.302421197	3.446020062
53	0.064763	30	30	0.927507	0.994707451	31.25763597
45	0.064763	8	8	0.927507	0.999999846	10.77667927
37	0.064763	-5	5	0.231877	0.304369431	4.05680457
29	0.064763	-7	7	0.927507	0.999999846	8.869808556
20	0.064763	-17	17	0.927507	0.999999846	17.46239985
6	0.064763	-15	15	0.927507	0.999999846	14.4222199
6	0.064763				0.072492626	0.411694053
7	0.064763	-39	39	0.927507	0.999999846	36.83976035

weight ave divided by overall signif	ave as quoted in Table S3	ave abs values
47.71764978	61	61.2
22.16063394	54	60.4
11.39477028	47	59
31.42394876	42	26.5
10.77668093	39	45
13.32855458	38	46.8
8.869809921	34	42.6
17.46240254	9	39.6
14.42222212	-2	14.2
5.679116312	-9	10
36.83976602	-24	21.2

Review of:

Greenland Ice Sheet runoff reduced by meltwater refreezing in bare ice

Submitted to: *Nature Communications*

Reviewer: Benjamin Hills

Summary.

Cooper et al. observe that meltwater runoff measured in the Greenland ablation zone is lower than would be predicted by climate models. They hypothesize that the reduction in runoff is associated with meltwater retention and refreezing in a near-surface layer of weathered ice, much like the relatively well-studied water retention in firn (Harper et al., 2012). They test their hypothesis with a pair of 1-dimensional thermodynamic numerical models they developed. They find that the model which considers light penetration and subsurface melting is a better representation for their measurements of meltwater runoff. The article is well written and a great fit for *Nature Communications* after the minor revisions I suggest below.

General Comments.

In my opinion, the difference in model output between SkinModel and IceModel is one of the more important results in this body of work. If you agree, I believe that some reorganization of the Results section could help emphasize that result to all readers. Currently, *Results* is broken out into subsections on: fieldwork, model overestimation of meltwater (SkinModel), explanation of overestimation (IceModel), upscaling to SW Greenland. The separation between explanations of SkinModel and IceModel could make it hard for the reader to appreciate what you are doing here (even though I do like your clarifying statement on L137-139). I would consider reorganizing as follows:

- Intro (pretty much as you have it)
- Results (field results only)
 - o You could consider moving Figs 2 and 3 up here, although I understand why you want to lead with what you now have as Fig 1.
- Analysis
 - o Bring your explanation of SkinModel and IceModel
 - o Compare to climate model output at the local scale (currently Fig 1)
 - o Compare to climate model output at the regional scale (currently Fig 4)
- Discussion/Conclusions (pretty much as you have it)

Perhaps this organization does not work well for NatureCommunications but that is how I would write it to emphasize your two models and the important difference between them (take it or leave it).

My second large takeaway from your article is the difference between the local scale (and supraglacial discharge) as opposed to regional scale (and proglacial discharge). In Figure 1 we see that the measurements are at the lower end of the model spread, whereas in Figure 4 the

measurements are closer to the middle of the spread (especially in c). There are several important points which come to mind for me which I do not think were mentioned in the article. First is the fundamental difference in the catchments. The physical processes important for your surface discharge measurements at RB will be entirely captured by your measurements and modeling efforts (as far as I can tell), but the proglacial discharge measurements have water that has moved through englacial and subglacial environments which you do not consider. Could basal melting provide additional component to discharge (you may be able to rule that out with a sentence or two)? What about groundwater? Could there be some groundwater flux not accounted for in the discharge measurements? I also wonder whether the uncertainty in defined catchment areas are sufficiently captured (I know they came from a different study). Compared to your surface catchment at RB, the regional catchments are more difficult to constrain with confidence and if subglacial water gets routed in unexpected ways it could elevate (or suppress) the measured proglacial discharge. I don't think this discredits your study, but worth mentioning some of these uncertainties.

I am generally confused by the use of the term "emulator". That term is often used to describe the statistical representation of a complex physical model which is highly computationally expensive (e.g. Conti et al., 2009; Gu et al., 2018). Here, I believe you are running the forward models (IceModel and SkinModel) directly. By that I mean that you are actually calculating the numerics on your prescribed 1-D mesh (which would not be done for an emulator). If I am mistaken, please correct me and add some context in the methods on the statistical emulation.

The supplementary material is referenced many times in the main article. I appreciate the thoroughness of the authors here, but so many references can disrupt the readability of the article. My personal opinion is that only the most important references to the supplementary material should be used, and that any figures which are critical to explain the main narrative should be included in the article itself (see my comment under Figures below).

Specific Comments.

L21 – Currently not clear whether the statement starting with “From 2009-2018...” is based on your work. I would start with something like “We found that, from 2009-2018, meltwater...”

L24 – Same as above, I believe this statement could be more clear if you are explicit that it is based on prior work, something like “The mass retention explains [the evidence from prior studies] that runoff could be overestimated...”

L43 – Some readers may not immediately understand that SW Greenland is the same “ablation zone” that you had been talking about above.

L56 – is “substrate” the correct word here?

L65 – I would remind the reader again with “...this hypothesis [of retention and refreezing]...”, mostly because folks who skim the article may jump to this paragraph.

L76 – Is “RB” defined somewhere? This is the first time I see it.

L86 – Somewhat confused by definition of runoff. Discharge/area would give units of m/s but you discuss runoff in Gt/a in the abstract and plot cumulative runoff in m³ (e.g. in fig 1).

L106 – “To investigate these discrepancies...”? explain makes it sound like you knew the answer coming into the study.

L131 – You could just cite your zenodo doi for IceModel instead of the supplement.

L137-139 – This is one of the most important sentences in my opinion, and I fear it may get lost in the current organization. See my general comment on organization of the results section above.

L147 – Not immediately clear why “July”? I would say “summer months” or “during the time of the field investigation” or similar.

L167 – Are you truly applying your models to “all bare ice areas of SW Greenland”? or just to the LG and AK4 catchments?

L244 – Why two standard deviations? One would be more typical and I think appropriate here. Either way, it is obviously going to be more certain than the models which is true even for two.

L306 – In the caption of supplemental figure S2 you give the total area of the stake survey as ~0.1 km². I believe that is a relevant number to include in the main document, could be in this paragraph which is why I mention it here.

L330 – Giving water density to 5 significant digits is implying a certainty that you do not have, but I do not feel strongly about changing it (certainly changing to 1000 will not change your analysis at all).

L380 – I am assuming that the “flotation factor” is an input to the GRASS algorithm? Would be useful to state that.

L434 – Can you strengthen the statement by dropping “to the best of our knowledge”? Seems to me like this list of authors would know if it were otherwise.

L478 – I am assuming that there are many 1-d models run in parallel? All with different surface boundary conditions? Might be worth clearing that up unless it is already clear and I missed it.

L502 – “5 km horizontal grid spacing”. Should be made clear (as in previous comment) that this is not a model mesh spacing (i.e. including horizontal energy transfer) but parallel 1d models. I don’t think it matters as the horizontal temperature gradient would be negligible, but worth making clear what you did.

L546 – I had trouble with the Leverett Glacier doi:

<https://doi.org/10.5285/17c400f1-ed6d-4d5a-a51f-aad9ee61ce3d>

but its possible that the BAS server was having issues when I checked. Anyway, double check me.

L551 – it seems like the PROMICE data portal may have changed to:

<https://promice.org/download-data/>

Figures.

I like Figure S2. I think it could elevate figures in the main article. That is, consider including (a) in Fig 1 of the main article and (b-d) in Fig 3 of the main article. If not this, then you should do *something* to make clear to the reader that Fig 1 is local scale compared to Fig 4c and 4d.

References

- Conti, S., Gosling, J. P., Oakley, J. E., & O'Hagan, A. (2009). Gaussian process emulation of dynamic computer codes. *Biometrika*, *96*(3), 663–676. <https://doi.org/10.1093/biomet/asp028>
- Gu, M., Wang, X., & Berger, J. O. (2018). Robust Gaussian stochastic process emulation. *Annals of Statistics*, *46*(6A), 3038–3066. <https://doi.org/10.1214/17-AOS1648>
- Harper, J. T., Humphrey, N. F., Pfeffer, W. T., Brown, J., & Fettweis, X. (2012). Greenland ice-sheet contribution to sea-level rise buffered by meltwater storage in firn. *Nature*, *491*, 240–243. <https://doi.org/10.1038/nature11566>

This manuscript uses field observations and surface energy balance modeling to demonstrate potential liquid water storage within west Greenland ice sheet bare ice. The results suggest that overestimation of runoff by regional climate models and the MERRA-2 global reanalysis may likely be attributable to storage and refreezing of water within the bare ice. These are important findings and hold potential for improving estimates of ice sheet mass change.

I've evaluated the manuscript as well as the previous second reviewer's evaluation and the authors' response. In general, I find the manuscript to be well written and supported, though I have a few concerns documented below that should be addressed before the manuscript should be considered further for publication.

Specific comments:

Regarding **R2's** point:

"Fig S11 shows that for areas of the SW sector above c.900m ..., RACMO albedo on average is actually higher than MAR albedo. ... So differences between RACMO and MAR are significant and not all about different albedo. This starts to mess up the assumptions held at the smaller scales above, and indicates processes other than albedo and refreezing may explain differences."

And the **authors** response:

"Fig. S11 (revised Fig. S10) does not support the reviewer's claim"

- **My take:** When I look at S10 (former S11) panels a and b that show the whole SW sector, I tend to agree with R2. For example, at 1500m, the MAR albedo trend line is ~0.66, whereas RACMO is ~0.71. As such, RACMO albedo is higher than MAR, on average, across the SW basin. This differs from the LG catchment (S10c) and RB catchment (S8), where RACMO has lower albedo than MAR. This then supports R2's claims that assumptions at the smaller scale regarding attributing differences in the models' runoff to albedo and refreezing may not apply to the entire SW basin. I don't think this detracts from the main points of the paper, but it should be acknowledged (in particular in the text ranging from L123-L134).

Additional concerns:

L107: The authors cite Fig S4-S5 to claim an evaluation of albedo in various models. However, these figures do not show albedo directly, but rather SW_{\uparrow} . Since albedo is the ratio of $SW_{\uparrow}/SW_{\downarrow}$, SW_{\uparrow} alone is insufficient to assess whether albedo is accurately

represented. For example, we don't know whether discrepancy in SW_{\uparrow} is a function of the model's albedo or rather, the amount of SW_{\downarrow} that is impacted primarily by the atmosphere. To support claims about fidelity in assessing albedo, the authors should present both SW_{\uparrow} and SW_{\downarrow} to compute albedo or revise their text to accurately state they are not directly evaluating albedo.

This issue is particularly evident in the authors' claim that RACMO closely reproduces observed albedo, citing Figure S4. However, a closer examination of this figure reveals that although RACMO represents SW_{\downarrow} and R_{net} reasonably well, the agreement is likely not because RACMO is getting *albedo* correct. Specifically, RACMO significantly underestimates LW_{\downarrow} during a multi-day period ~July 9-11, suggesting it underrepresents cloudiness. Despite this, RACMO matches R_{net} , likely through a compensating error – because it has similar SW_{\uparrow} and seemingly misses this cloudy period (evidenced through lack of LW_{\downarrow}), the only way it can match R_{net} is by having a lower albedo than the surface observations. In other words, more SW would be incoming due to lack of clouds in RACMO, and to match the SW_{\uparrow} and R_{net} , it must be absorbing significantly more SW radiation and thus have a lower albedo. It gets the right answer for the wrong reasons.

This lack of critical evaluation raises **significant concern** that the authors have not closely considered the data upon which they are basing their statements. At minimum, I would suggest the authors should reexamine their evaluation framework and revise overly strong claims that are not directly supported by the figures or data presented.

L108: Terminology of “net radiative heat fluxes” is strange – should probably say “net radiation” or “turbulent heat fluxes and net radiation”. Seems like maybe this was a consequence of word cutting so the assessment of the turbulent heat fluxes and net radiation got merged.

L129: Again, the authors have not demonstrated that RACMO has the most accurate albedo. Revise.

L148 / Figure S9: Monthly labels in the inset figure are very hard to interpret – for example which line relates to which month is almost impossible to assess. I would suggest the authors consider revising this figure to potentially include using a legend that links the line color/style or month number.

Figure 4: Referring to panel b, the ice core density profile suggests that the subsurface is even more porous than IceModel, and thus there's even more potential for storage of meltwater. I am curious how the authors have accounted for this discrepancy in their model or results.

L182 / Figure 5: Panels c and d both show 8 lines each that are very difficult to differentiate. Part of this is the small size of each subplot and part of this is that the weight of the lines and dash patterns in the legend do not equal the weight and patterns of the lines in the actual plots. For example, which green line is RACMO and which is IceModel? Which dashed/dotted line is SkinModel and which is MERRA-2?

L186 (and elsewhere): This is a stylistic thing, but “underpredicts by -15%” is a double-negative. Suggest using the directional verb or signed value, not both.

L406 onward (Method section related to climate models):

- I suggest the authors should carefully assess which modeled products they used and whether the citations here are correct. For example, the authors state they used RACMO2.3p3, citing reference 22 (Noël 2018). The Noël paper discusses RACMO2.3p2 (not p3).
- Beyond this, I am concerned about inconsistencies in the RCMs used in this study. MAR data are ~15km resolution and forced with ERA5. The RACMO data are apparently ~11km and forced with ERA-Interim. Ideally, the authors should be using more recent RACMO data forced by ERA5 for a more fair assessment of the relative ability of MAR and RACMO.
- The authors state they use both RACMO2.3p3 and RACMO2.3p2. The citation for the p2 data is for the 1km downscaled version of RACMO2.3p2, yet the authors state they used ~11km RACMO products.
- Both MAR and RACMO have been statistically downscaled to ~1km resolution. Why are the authors not using these higher resolution products that would in theory better represent catchment-scale meltwater processes?